# An osteoinductive and biodegradable intramedullary implant accelerates bone healing and mitigates complications of bone transport in male rats

Sien Lin [1,7], Hirotsugu Maekawa[1,7], Seyedsina Moeinzadeh[1], Elaine Lui[1,2], Hossein Vahid Alizadeh[1], Jiannan Li[1], Sungwoo Kim[1], Michael Poland[3], Benjamin C. Gadomski[3], Jeremiah T. Easley[4], Jeffrey Young[1], Michael Gardner[1], David Mohler[1], William J. Maloney[1] & Yunzhi Peter Yang [1,5,6] ✉

Bone transport is a surgery-driven procedure for the treatment of large bone defects. However, challenging complications include prolonged consolidation, docking site nonunion and pin tract infection. Here, we develop an osteoinductive and biodegradable intramedullary implant by a hybrid tissue engineering construct technique to enable sustained delivery of bone morphogenetic protein-2 as an adjunctive therapy. In a male rat bone transport model, the eluting bone morphogenetic protein-2 from the implants accelerates bone formation and remodeling, leading to early bony fusion as shown by imaging, mechanical testing, histological analysis, and microarray assays. Moreover, no pin tract infection but tight osseointegration are observed. In contrast, conventional treatments show higher proportion of docking site nonunion and pin tract infection. The findings of this study demonstrate that the novel intramedullary implant holds great promise for advancing bone transport techniques by promoting bone regeneration and reducing complications in the treatment of bone defects.

Segmental bone defects caused by high-energy trauma, debridement procedures, or tumor resection remain great challenges in the field of orthopedics[1-3]. Limb salvage techniques such as fracture stabilization or reconstruction have been applied to manage large bone defects for decades[4,5]. In particular, the Ilizarov technique, based on the principles of distraction osteogenesis (DO), represents a well-established limb salvage procedure for managing large bone defects[6,7]. This procedure starts with a low energy osteotomy, followed by lengthening over an external or internal fixator[6,8]. Ilizarov described two distinct techniques: acute shortening and subsequent lengthening, and bone transport[9-13]. Bone transport is used due to its advantages in the treatment of larger bone defects without limb discrepancy and soft tissue contraction[10,12]. However, it also has several disadvantages, including pin tract infection, docking site nonunion, and poor bone regeneration, which may prolong the treatment duration[14]. Among these, docking site nonunion is frequently encountered due to inactive

[1]Department of Orthopaedic Surgery, School of Medicine, Stanford University, Stanford, CA 94305, USA. [2]Department of Mechanical Engineering, School of Engineering, Stanford University, Stanford, CA 94305, USA. [3]Orthopaedic Bioengineering Research Laboratory, Department of Mechanical Engineering, Colorado State University, Fort Collins, CO 80523, USA. [4]Preclinical Surgical Research Laboratory, Department of Clinical Sciences, Colorado State University, Fort Collins, CO 80523, USA. [5]Department of Materials Science and Engineering, School of Engineering, Stanford University, Stanford, CA 94305, USA. [6]Department of Bioengineering, School of Medicine, Stanford University, Stanford, CA 94305, USA. [7]These authors contributed equally: Sien Lin, Hirotsugu Maekawa. ✉e-mail: ypyang@stanford.edu

bone contact and soft tissue intrusion at the docking site which can lead to pseudarthrosis[12,15]. To achieve bone bridging at the docking site, secondary debridement and bone grafting surgery are often necessary, causing additional pain, morbidity, hospitalizations, and economic burden for patients. Therefore, a single-surgery approach to treat bone defects and avoid secondary surgeries is desired.

Recently, there has been an increasing number of reports on using metallic intramedullary (IM) nail in bone transport procedure to reduce the use or period of external fixation[16–18]. These IM implants are typically made of titanium alloys and have been routinely used for internal fixation in orthopedic surgery for several decades[19–22]. While metallic IM implants have advantage such as higher patient satisfaction and less social-psychological concerns[19], there is no reliable evidence

to suggest that they achieve early consolidation or reduce complications[23], and an additional surgery may be needed to remove the metallic IM nail[19,24]. Biodegradable IM nails offer an alternative to avoid potential secondary removal surgery. As early as 1992, a biodegradable IM nail made from polyglycolic and polylactic acid co-polymers was developed for IM fixation and showed no significant difference in bone healing compared to the control group using Kirschner wires for the fixation of extraarticular fracture[25]. Antibiotic eluting IM implant apparatuses were later developed for fracture fixation or other bone reconstruction procedures, as open wounds are exposed to bacterial or other infectious micro-organisms, showing a promising effect in reducing infection and promoting bone healing[26–28]. A recent animal study reported that a biodegradable

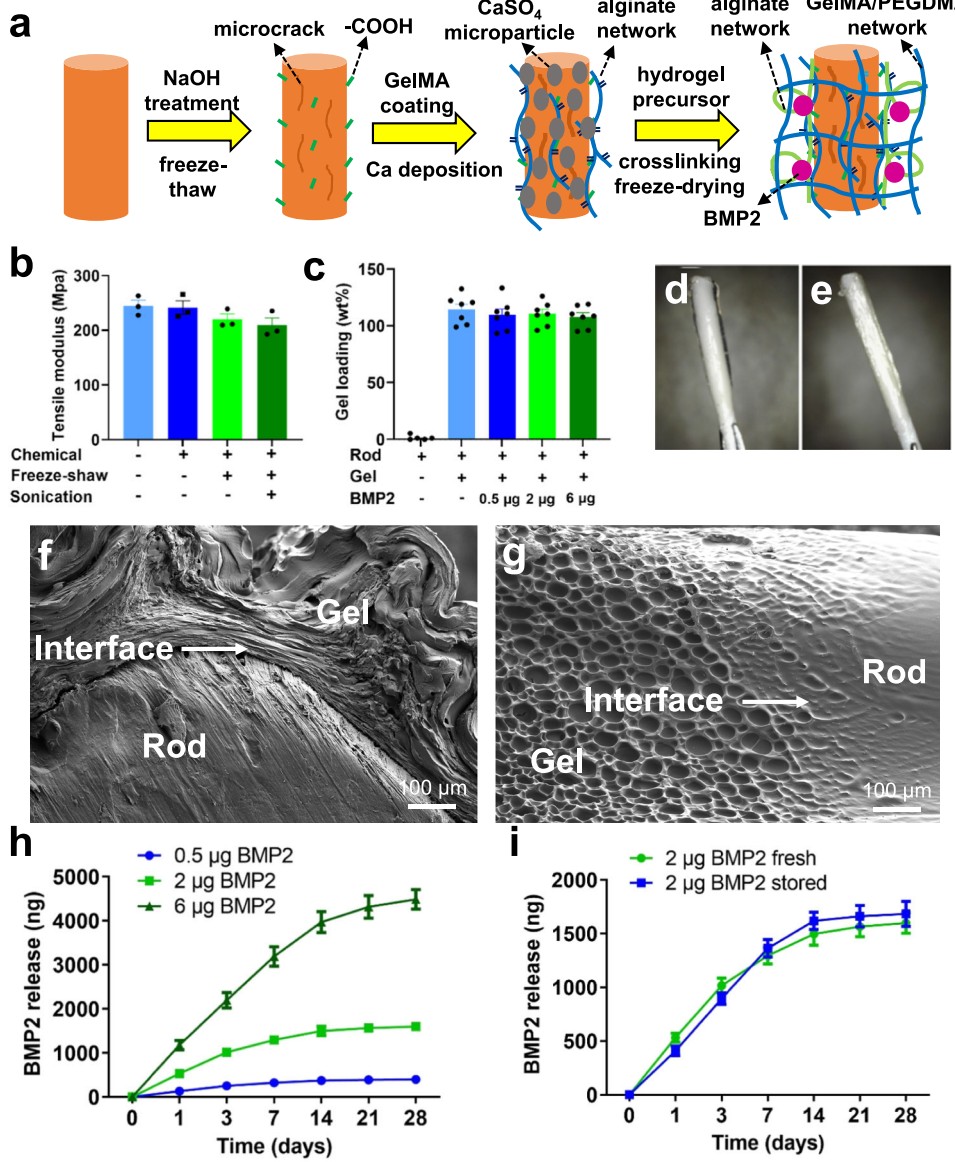

**Fig. 1 | Fabrication and characterization of a bone morphogenetic protein-2 (BMP2) eluting poly-caprolactone/beta-tricalcium phosphate (PCL-TCP) implant. a** Schematic illustration of the HyTEC technique. After extrusion, PCL-TCP filament was consequently treated with chemicals, freeze-thaw, sonication, and hydrogel coating. **b** Tensile modulus of the filaments measured after different treatments (*n* = 3 independent samples per group). **c** Hydrogel loading efficiency of the filaments with different amounts of BMP-2 (0, 0.5 µg, 2 µg or 6 µg, *n* = 7 independent samples per group). **d, e** Photographs of a wet (**d**) and a freeze-dried (**e**) implant after coating. **f** Scanning electron microscopy (SEM) images of hydrogel-

PCL-TCP filament interface from cross-sectional view. **g** SEM images of hydrogel-PCL-TCP filament interface from longitudinal view. **h** BMP-2 release profile in the implants loaded with 0.5 µg, 2 µg or 6 µg BMP-2 over 28 days (*n* = 3 independent samples per group). **i** BMP-2 release profile in the fresh-made or stored implants loaded with 2 µg BMP-2 over 28 days (*n* = 3 independent samples per group). Data are presented as scatter dot plot or curves with mean and s.e.m. values. This ex vivo characterization was repeated at least two times independently. Source data are provided as a Source data file.

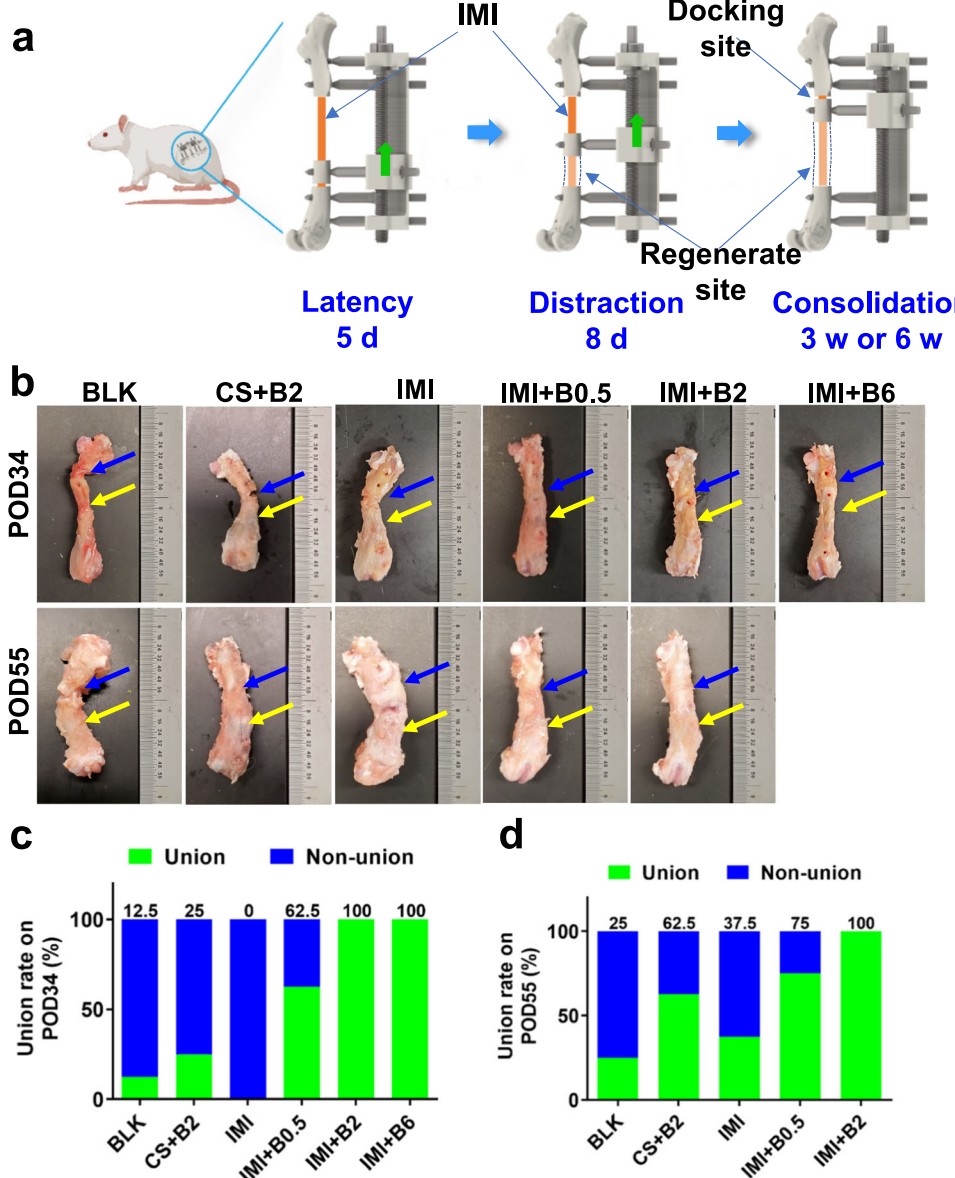

**Fig. 2 | Animal experimental design and bony union assessment. a** Schematic illustration of femoral bone transport over the IM implant (IMI) in a rat femur. **b** Photographs of the femoral specimens harvested on post operation day 34 (POD34) or POD55. The rats were subjected to bone transport only (BLK) or implantation of collagen sponge (CS) preabsorbed with BMP-2 (2 μg). Other rats were treated with IM implants only (IMI) or implants which were loaded with 0.5 μg (IMI + B0.5), 2 μg (IMI + B2) or 6 μg (IMI + B6) BMP-2. Blue and yellow arrows point to the docking sites or regenerate sites. **c** Bony union rates on POD34 determined by imaging (*n* = 8). **d** Bony union rates on POD55 (*n* = 8). Union rates data are presented as percentages. Source data are provided as a Source data file.

magnesium IM nail promoted consolidation in bone lengthening by promoting angiogenesis[29]. However, until now, there have been no reports of patient or animal models in which bone transport was treated with a biodegradable IM implant.

To achieve early bone consolidation, osteoinductive growth factors such as bone morphogenetic protein-2 (BMP-2) have been shown to significantly promote bone defect healing[30]. BMP-2 and BMP-4 expression is high in the the regenerated tissue during the distraction phase and then gradually decreases during the consolidation phase, indicating that supplementary doses of BMP-2 during the consolidation phase may be beneficial[31,32]. Local injection of recombinant human (rh) BMP-2 or BMP-7 can accelerate bone formation in DO animal models[33–36]. However, there is a lack of sustained delivery methods for BMP-2 during DO. To address this unmet need, a hybrid tissue engineering construct (HyTEC) technique was developed to enable sustained delivery of BMP-2 from a biodegradable IM implant as an adjunctive therapy of bone transport in a single procedure to accelerate consolidation and avoid secondary operations, grafting, and other severe complications. More specifically, the goal of this research was to address clinical challenges, such as prolonged consolidation and high nonunion rates in bone defect healing, through a unique bioengineering solution, an osteoinductive biodegradable IM implant device. To this end, a clinically relevant rodent model of bone transport was established, and a novel BMP-2 eluting biodegradable IM implant device was developed and examined for sustained-release ex vivo. The effect of the novel BMP-2-eluting IM implants on bone consolidation and docking site union was then investigated in the bone transport rodent model. The BMP-2-eluting IM implant device is made of Food and Drug Administration (FDA)-cleared growth factor and materials. This study aims to provide a solution for a single treatment that combines a clinically available surgical approach with a unique bioengineering solution.

## Results

### Characterization of BMP-2 eluting IM implants

The HyTEC technique for fabricating an osteoinductive biodegradable IM implant is schematically shown in Fig. 1a. Following extrusion of polycaprolactone-beta-tricalcium phosphate (PCL-TCP, 4:1 in w/w) filaments, the BMP-2-laden hydrogel was loaded onto the surface of the filaments (Fig. 1a). Before hydrogel coating, the surface of the filaments was treated with NaOH, frozen, and then thawed to increase the roughness and microporosity (Fig. 1a and Supplementary Fig. 1a, b), and then deposited with $CaSO_4$ microparticles as crosslinkers (Supplementary Fig. 1c). A porous BMP-2-laden hydrogel layer was formed on the surface of the filaments by surface-initiated physical crosslinking of alginate followed by covalent crosslinking of GelMA and PEGDMA, and the construct was then freeze-dried (Supplementary Fig. 1d–f). Mechanical tests showed no significant difference in the tensile modulus in any of the treated filaments and the untreated ones (Fig. 1b). Hydrogel coating increased the wet weight (Fig. 1c). The hydrogel layer was attached firmly to the filament after freeze-drying, showing a smooth transition from the hydrophobic filament to the hydrophilic hydrogel at the interface (Fig. 1d–g, Supplementary Figs. 1e, f and 2) and robust adhesion strength, with an average interfacial stiffness equaling of (0.609 ± 0.194) MPa (Supplementary Fig. 3). BMP-2 was released from the freeze-dried hydrogel-loaded implant in a sustained and dose-dependent manner over 21 days (Fig. 1h). BMP-2 released from the stored implants stored for 2 months at 4 °C showed similar release kinetics to the freshly prepared implants (Fig. 1i), indicating a minimum extended shelf-life of at least 2 months. A prolonged release of BMP-2 could be achieved by coating the IM implant with additional layers of PCL. The amount of BMP-2 released within 28 days reduced from 84% in the original IM implant to 62% or 24% when coated with one or three layers of PCL, respectively (Supplementary Fig. 4). This modified HyTEC (mHyTEC) technique using a multilayer coating approach could be used for a prolonged release in future studies involving large animal models and humans. These studies indicate that with the HyTEC technique, a unique surface coating technique can incorporate a broad range of BMP-2 dosages on the surface of the scaffolds and can allow for a tunable sustained release kinetics.

### Imaging and mechanical evidence of functional bone regeneration

The efficacy of BMP-2-laden IM implants was tested in a rat bone transport model (Fig. 2a). The grouping methods and related assessments are shown in the Table S1. A novel monolateral external fixation device was specifically designed and customized for the in vivo experiments (Supplementary Figs. 5 and 6). Results showed that the intercalary segment could be transported smoothly over an IM implant, keeping an intact freeze-dried coating during bone transport (Supplementary Figs. 5 and 6). The bone transport procedures consisted of 5-day latency, 8-day distraction, and 3-week or 6-week consolidation (Fig. 2a). The intercalary segment was transported retrogradely over the IM implant during 8-day distraction (Fig. 2a). Bone healing could be monitored by X-ray imaging (Supplementary Fig. 7). At the endpoints, which were post operation day 34 or 55 (POD34 or 55), the blank surgical control (BLK) group showed a lower union rate at the docking site (12.5% and 25% on POD34 and POD55, respectively, Fig. 2b–d), suggesting a clinically relevant animal model. The BMP-2-laden (2 µg) collagen sponge (CS + B2) group showed an improvement in the union rate to some extent (Fig. 2c, d), whereas the IM implant (IMI) group did not show any improvement in terms of union rate (Fig. 2c, d). Encouragingly, the BMP-2-laden IM implants showed higher union rates. In the 0.5 µg BMP-2-laden IM implant group (IMI + B0.5), the union rate increased from 62.5% (POD34) to 75% (POD55) (Fig. 2c, d). More surprisingly, all the femoral samples showed 100% union in the 2 µg BMP-2-laden IM implant group (IMI +

B2) and 6 µg BMP-2-laden IM implant group (IMI + B6) (Fig. 2c, d). Of note, there was only one timepoint (POD34) in the (IMI + B6) group.

Micro-CT analysis was performed to reconstruct the 3D images of the femurs and to determine the quantitative bone mass in the two regions of interest (ROIs): the regenerate site and the docking site, with bone volume/tissue volume (BV/TV) normalized to the contralateral intact bone before comparison. Overall bone mass continued to increase over time in all groups. Notably, the (IMI + B2) group showed significantly higher bone mass at the docking site (+34.7% in BV/TV, $p = 0.0015$ on POD34; +55.9% in BV/TV, $p = 0.0001$ on POD55) and the (IMI + B6) group also showed an increase (+39.7% in BV/TV, $p = 0.0002$ on POD34) compared to the BLK group (Fig. 3a–c, Supplementary Figs. 8 and 9). Compared with the IMI group, the (IMI + B2) group showed a significant increase in BV/TV by 28.6% ($p = 0.0128$) or 51.2% ($p = 0.0004$) at the docking site on POD34 or POD55, and the (IMI + B6) group showed a 33.5% increase ($p = 0.0023$) on POD34 (Fig. 3b–c). There was also significantly higher BV/TV in the (IMI + B2) group at the docking site on POD55 compared to the (CS + B2) or (IMI + B0.5) groups. Newly formed bone was found at the regenerate site in all the groups with no significant difference in BV/TV on POD34. Bone mass in the regenerate site was significantly increased only in the (IMI + B2) group (+31.2% in BV/TV, $p = 0.0205$) compared to the BLK group on POD55 (Fig. 3c), with no significant difference between the (IMI + B2) and (IMI + B6) groups on POD34 (Fig. 3b). In summary, µCT data indicate that IM implants incorporating 2 µg or 6 µg BMP-2 significantly enhance bone regeneration at the docking sites.

Three-point bending mechanical tests were performed on femoral specimens harvested on POD34 (Fig. 3d–f) and POD55 (Fig. 3g–i) to determine their mechanical properties. For the nonunion samples, the value of mechanical parameters was set to zero as their failure in testing. Mechanical properties increased over time in each group with the (IMI + B2) group showing significant enhancement in maximum load, Young's modulus, and energy absorption compared to the BLK group, increasing by 22.5% ($p = 0.0026$), 7.4% ($p = 0.0328$), or 16.8% ($p = 0.0475$) on POD34 (Fig. 3d–f), or 32.6% ($p = 0.0000$), 32.8% ($p = 0.0002$), or 17.8% ($p = 0.0073$) on POD55, respectively (Fig. 3g–i). Mechanical properties were also significantly enhanced in the (IMI + B2) group compared to the IMI group, increasing by 21.7% ($p = 0.0041$) or 7.0% ($p = 0.0466$) on POD34 (Fig. 3d–f), or 28.6% ($p = 0.0000$) or 33.5% ($p = 0.0002$) on POD55 (Fig. 3g–i) in maximum load or Young's modulus, respectively. Furthermore, maximum load was significantly increased in the (IMI + B2) group on POD55 compared to the (CS + B2) group (Fig. 3g–i). Similar results could be found in the (IMI + B6) group (Fig. 3d, e). Gait performance was investigated in the IMI and (IMI + B2) groups ($n = 2$) on POD48 (with fixator) and POD55 (fixator removed on POD48). We found that all the rats could move freely on POD48 before fixator removal (Supplementary Movie 1 and 2). However, the rats in the IMI group had a limping and hobbling gait on POD55 after fixator removal (Supplementary Movie 3). Notably, the rats in (IMI + B2) group moved quickly and fluently on POD55 after fixator removal (Supplementary Movie 4), indicating a satisfied load bearing efficacy after 6 weeks of consolidation. In this study, although we could not compare the mechanical properties of the distal docking site and proximal regenerate, we found that almost all of the healed specimens were broken at the docking site at the ultimate load during testing. Furthermore, nonunion or delayed union could be only found at the docking site in the rats, which is consistent with clinical evidence[14].

### Histological evidence of bone regeneration

To provide direct evidence of bone regeneration at the tissue level, histological analysis was also conducted on the femoral samples (Figs. 4–6, Supplementary Figs. 10–15). Fibrous tissue interposition was observed at the docking sites of BLK, IMI, (CS + B2), and (IMI + B0.5) groups on POD34 and POD55 (Fig. 4 and Supplementary Fig. 10). However, complete bony fusion at the docking sites and higher

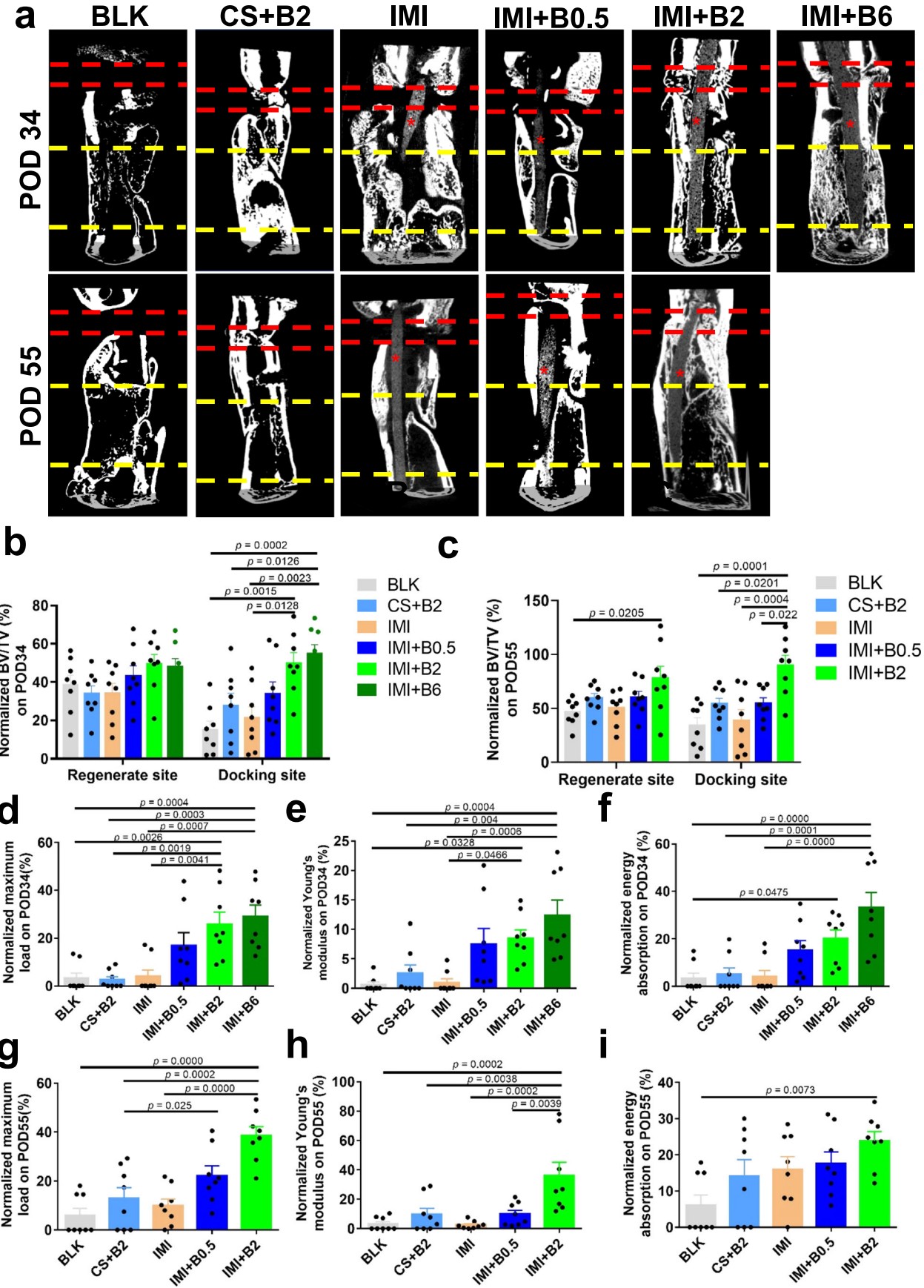

**Fig. 3 | Bone mass and mechanical properties of the femoral specimens. a** 2D micro-CT images showing bone regeneration at regenerate sites (between yellow dashed lines) and docking sites (between red dashed lines) with or without implants (asterisk) on POD34 and POD55. **b, c** Micro-CT quantification of bone volume/tissue volume (BV/TV) on POD34 (**b**) and POD55 (**c**) normalized by those of contralateral femurs. One-way ANOVA and Tukey's multiple comparisons test were applied in the analysis of micro-CT data ($n = 8$ independent rats per group). All imaging quantitative data are presented as mean ± SD. **d–f** Mechanical properties

measured by three-point bending tests in femoral specimens harvested on POD34, including maximum load (**d**), Young's modulus (**e**), and energy absorption (**f**). **g–i** Mechanical properties in femoral specimens harvested on POD55, including maximum load (**g**), Young's modulus (**h**), and energy absorption (**i**). Kruskal–Wallis test with Dunn's post hoc testing were applied in the analysis of mechanical testing results ($n = 8$ independent rats per group). Data are presented as scatter dot plot with mean and s.e.m. values. Source data are provided as a Source data file.

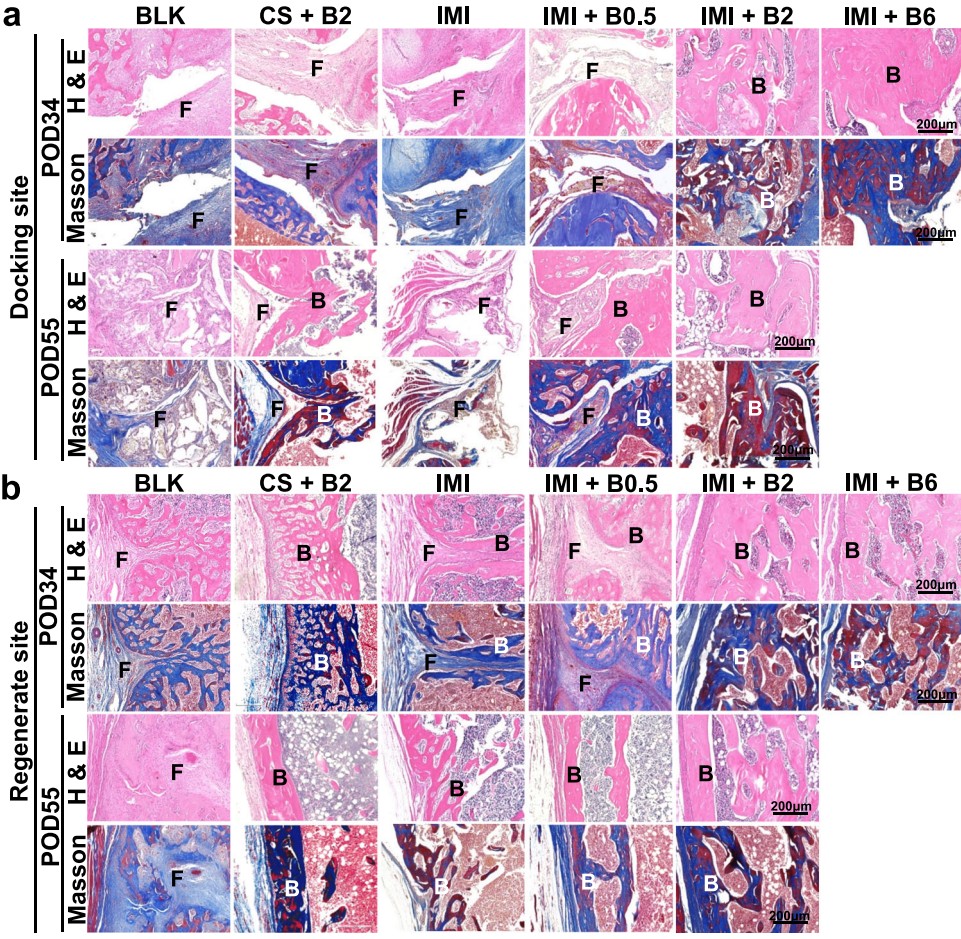

**Fig. 4 | Histology of bone regeneration at docking sites or regenerate sites in the femoral specimens on POD34 or POD55. a** Representative microphotographs at docking sites. **b** Representative microphotographs at regenerate sites.

Longitudinal sections were stained by hematoxylin and eosin (H&E) or Masson trichrome. F fibrous tissue, B regenerated bone. Histological staining was conducted three times independently.

amounts of newly formed bone surrounding IM implant were observed in the (IMI + B2) and (IMI + B6) groups as early as POD34, without inflammatory appearance (Fig. 4, Supplementary Figs. 10 and 11). Large amounts of newly formed bone were consistently observed at the regenerate site in all the groups at both endpoints (Fig. 4 and Supplementary Fig. 10). These histological results demonstrate that IM implants incorporating 2 μg or 6 μg BMP-2 safely and effectively promote bony fusion at the docking sites without significant difference at the regenerate sites.

To provide insights into bone remodeling, in-situ expression of osteogenic or osteoclastic activity was measured by immunohistochemistry (osteocalcin, OCN) and tartrate-resistant acid phosphatase (TRAP) staining, respectively. Notably, the expression levels of OCN were increased dose-dependently at the docking sites in the groups treated with BMP-2-laden IM implants on POD34 (Fig. 5a and c). Similarly, a higher number of osteoclasts per bone surface (Oc.N/BS) was observed at the docking sites in the (IMI + B2) and (IMI + B6)

groups on POD34 (Fig. 5b and d), indicating a coupling between active bone formation, bone resorption, and early bone remodeling during bone healing. However, there was no significant difference in the osteoclastic activity at the regenerate sites among the groups on POD34 (Fig. 5b and Supplementary Fig. 12). In addition, the expression of BMP-2 also exhibited a dose-dependent increase at the docking site (Supplementary Fig. 13). Consistent with the results of osteogenic markers, the expression of VEGF as well as CD31 exhibited significantly higher expression in the BMP-2-laden IM implant groups at the docking sites on POD34 compared to the BLK or IMI group (Supplementary Figs. 14 and 15). At the regenerate sites, there was no significant difference in either VEGF or CD31 expression among the groups, except for the (IMI + B6) group on POD34 (Supplementary Figs. 14 and 15). In summary, the results indicate that osteogenesis, angiogenesis, and bone remodeling may be remarkably enhanced at early time points by sustained BMP-2 release at the docking sites but not at the regenerate sites.

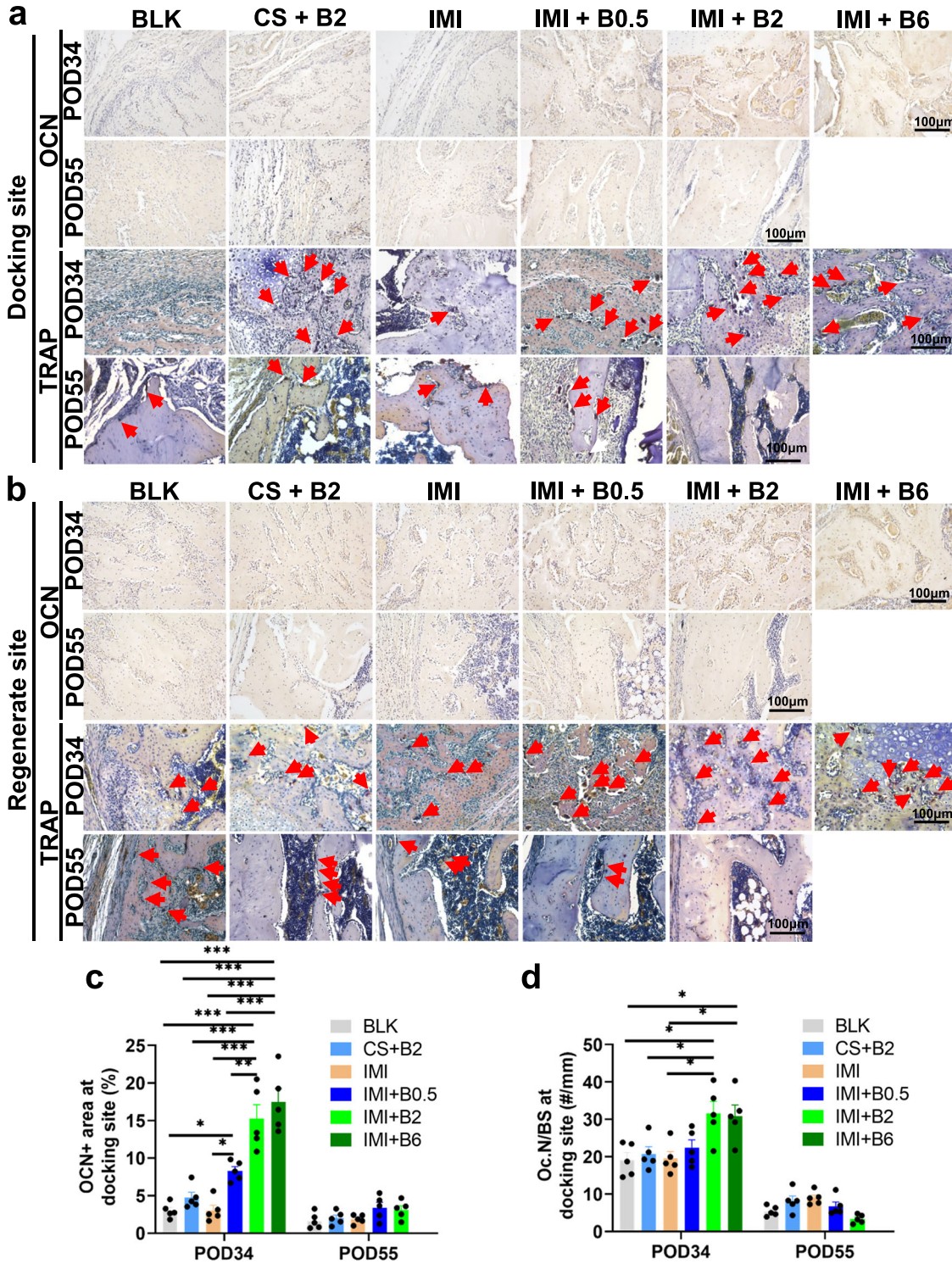

**Fig. 5 | Bone remodeling at docking sites or regenerate sites in the femoral specimens on POD34 or POD55.** Osteogenic marker (osteocalcin, OCN) or osteoclasts marker (tartrate-resistant acid phosphatase, TRAP) was stained by immunohistochemistry or TRAP assay. **a** Representative images of bone remodeling at docking sites. Arrows indicate TRAP-positive osteoclasts. **b** Representative images of bone remodeling at regenerate sites. **c** Semi-quantitative results of OCN positive expression area at the docking sites ($n = 5$ independent rats per group). **d** Semi-quantitative results of TRAP-positive osteo-clasts numbers per bone surface (Oc.N/BS) at the docking sites ($n = 5$ independent rats per group). Data are presented as scatter dot plot with mean and s.e.m. values. One-way ANOVA and Tukey's multiple comparisons test. Source data are provided as a Source data file.

To get insight into dynamic bone formation during the investigating period, non-decalcified bone histomorphometry was conducted by measuring the distance between in vivo fluorescent labels. Mineral apposition rate (MAR) was significantly higher in the (IMI + B2) and (IMI + B6) groups at the docking sites on POD34 (Fig. 6a, b). There was no significant difference in MAR at the regenerate sites among the groups on POD34, except for the (IMI + B6) group, which had the highest MAR (Fig. 6a, b). However, there was no significant change in MAR at both the docking site and regenerate sites among the groups on POD55 (Fig. 6a and c). These results indicate that there is a

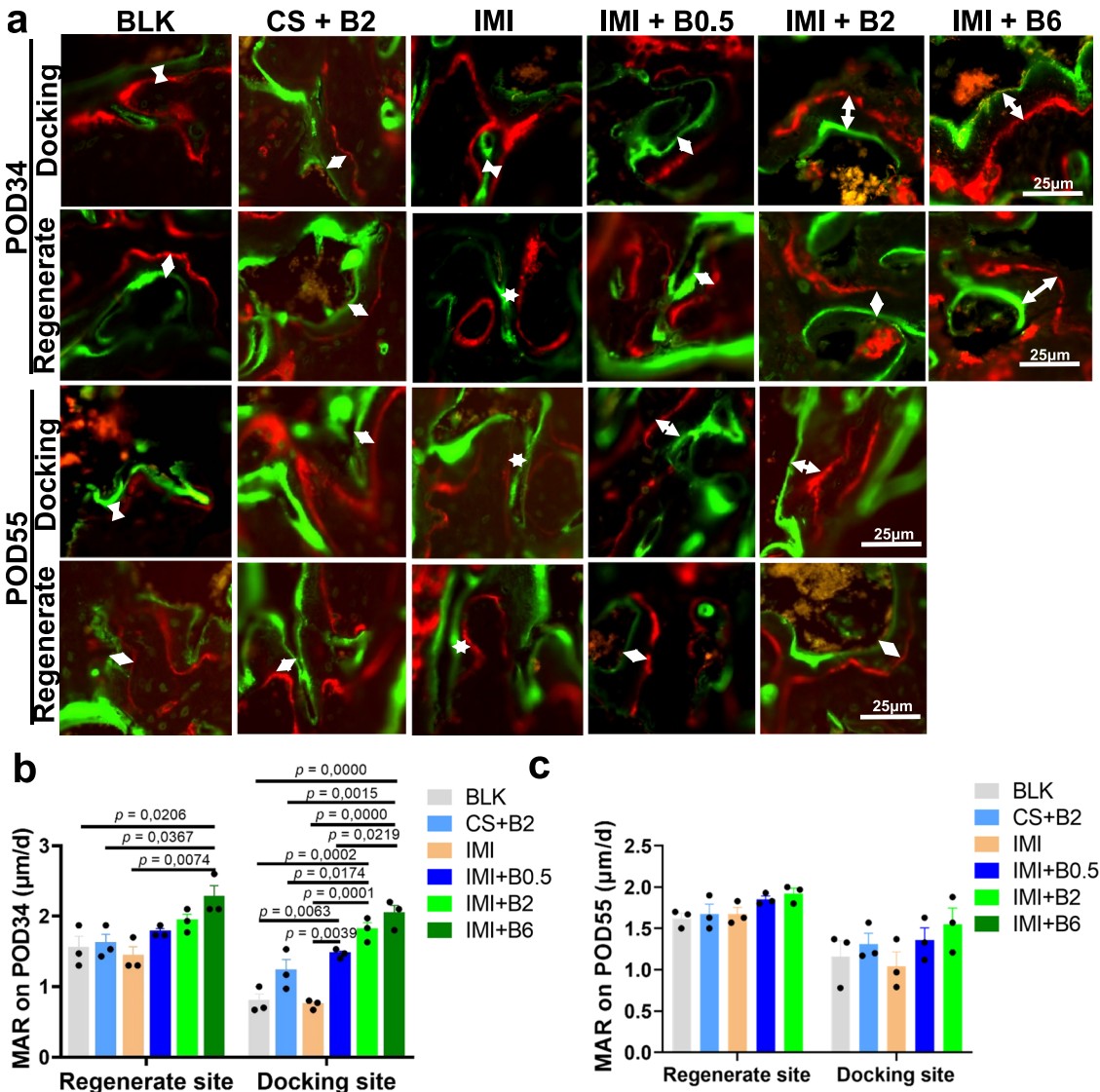

**Fig. 6 | Dynamic bone formation at docking sites or regenerate sites in the femoral specimens on POD34 or POD55. a** Representative fluorescent images of regenerated bone. The distance (white arrows) between green (calcein) and red (xylenol orange) fluorescent labels represents the dynamic bone formation in 10 days. **b** Quantitative results of mineral apposition rate (MAR) on POD34. **c** Quantitative results of MAR on POD55. Data are presented as scatter dot plot with mean and s.e.m. values ($n = 3$ independent rats per group). One-way ANOVA and Tukey's multiple comparisons test. Source data are provided as a Source data file.

dose-dependent, BMP-2-driven dynamic bone formation was active at the docking site in the early phases of bone healing.

## Pin tract infections and bacteria identification

Pin tract infection has a critical impact on bone regeneration when an external fixation device is applied. In this study, inflammation and enlarged pin tracts in some of the samples of the BLK, (CS + B2), and IMI groups were evidenced by histology and μCT analysis, indicating pin tract infection in those samples (Fig. 7a, Supplementary Figs. 16 and 17). Some animals in the BLK, (CS + B2), IMI, and (IMI + B0.5) groups showed minor infectious appearance, such as skin erythema or purulent discharge at pin tracts, with higher infection rates over time (Fig. 7b, c, Supplementary Fig. 16). However, there was no significant difference in the infection rate between BLK and IMI groups over time, suggesting that the IM implant does not cause additional infection during degradation. Interestingly, the (IMI + B2) and (IMI + B6) groups did not show any sign of infection (Fig. 7a, Supplementary Figs. 16 and 17 and Table S2). Bacterial colony formation assays showed higher proportions of positive bacterial culture in the IMI group than the

(IMI + B2) group on POD34 and POD55 in the pin samples (Supplementary Fig. 18). Further analysis using DNA next generation sequencing revealed that *Proteus* and *Morganella*, which are fecal origin bacteria, were more abundant in the IMI group, suggesting that these bacteria may invade along the loosening pin tracts and contribute to pin tract infections (Fig. 7d, Supplementary Fig. 19).

## Biological effect of eluting BMP-2 in early phase of bone transport

In addition to the relatively late endpoints, this study also examined the early phase of bone transport to investigate the molecular mechanism of eluting BMP-2. Rats were subjected to bone transport for 4 days without implant (BLK) or with an IM implant incorporated with 2 μg BMP-2 (IMI + B2), and a bone defect control group (CTRL) without bone transport or implant was treated with acute bone transport on the date of osteotomy (Fig. 8a). Active bone formation was observed at the distal site (regenerate) of the BLK group, as well as both the distal site (regenerate) and the proximal site (docking) of the (IMI + B2) group on POD13 (Fig. 8b). However, no bone formation

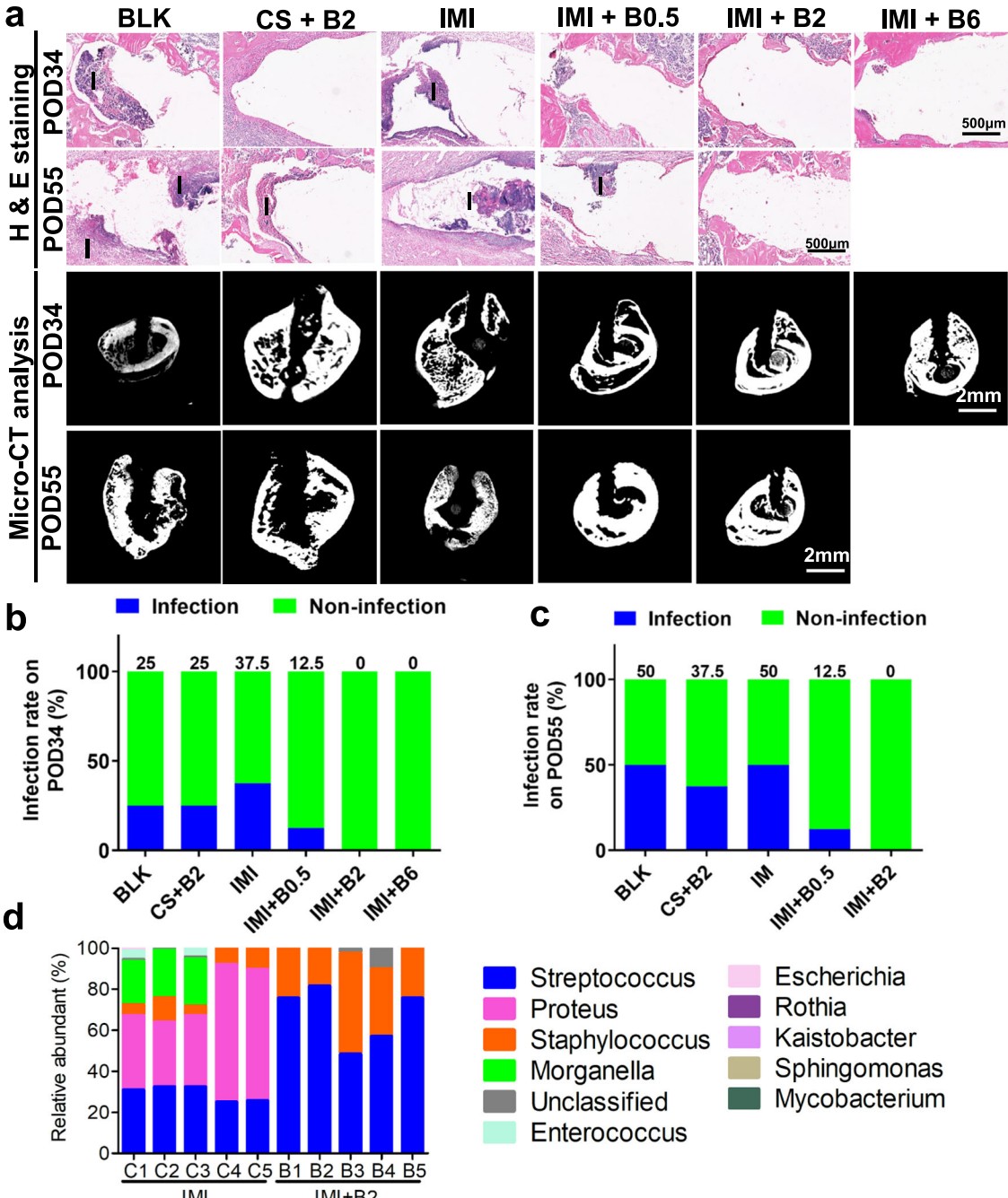

**Fig. 7 | Pin tract infections in the femoral specimens on POD34 or POD55.** **a** Histological (longitudinal section, H&E staining) and imaging (cross-sectional 2D micro-CT) results showing pin tract conditions. **b–c** Pin tract infection rate determined by visual inspection on POD34 (**b**) or POD55 (**c**). Infection rate data are presented as percentages, $n = 8$ independent rats per group. **d** Most abundant bacteria at genus level identified by 16 S next generation sequencing. Histological staining was conducted three times independently. Source data are provided as a Source data file.

could be found in the CTRL group. To further elucidate the molecular mechanism of DO and BMP-2, tissues at distal sites and proximal sites were collected for microarray analysis on POD13 (Fig. 8c–h and Supplementary Fig. 20). The regenerate site in BLK showed elevated expression of osteoblast/osteocyte markers (*Alpl, Ibsp, Dmp1*) and chondrocyte markers (*Acan, Col2a1, Col10a1*), suggesting that DO promotes active endochondral ossification during the early phase of DO (Fig. 8c–h). Eluting BMP-2 upregulated the expression of osteogenic markers (*Alpl, Dmp1, Ibsp*) and downregulated skeletal muscle markers (*Mb, Myog, Myoz1*) (Fig. 8c–h), suggesting that BMP-2 promoted bone regeneration but prevented skeletal muscle differentiation at the docking site. There was no significant difference in the

expression of osteoblast/osteocyte markers and MSC markers between BLK distal and (IMI + B2) proximal sites, but the expression of chondrocyte markers and TGFβ signaling were higher in BLK distal site, while the expression of BMP signaling was higher in (IMI + B2) proximal site (Fig. 8e–i). The combination of DO and eluting BMP-2 showed higher expression of osteoblast/osteocyte markers, osteoclast markers, and upregulated BMP and WNT signaling compared to each single treatment, suggesting that the combination of DO and (IMI + B2) has a synergistic effect on bone regeneration during the early phase (Fig. 8f–i). In summary, the combination of DO with eluting BMP-2 synergistically promoted bone formation via a different pathway that enabled early and robust bone formation.

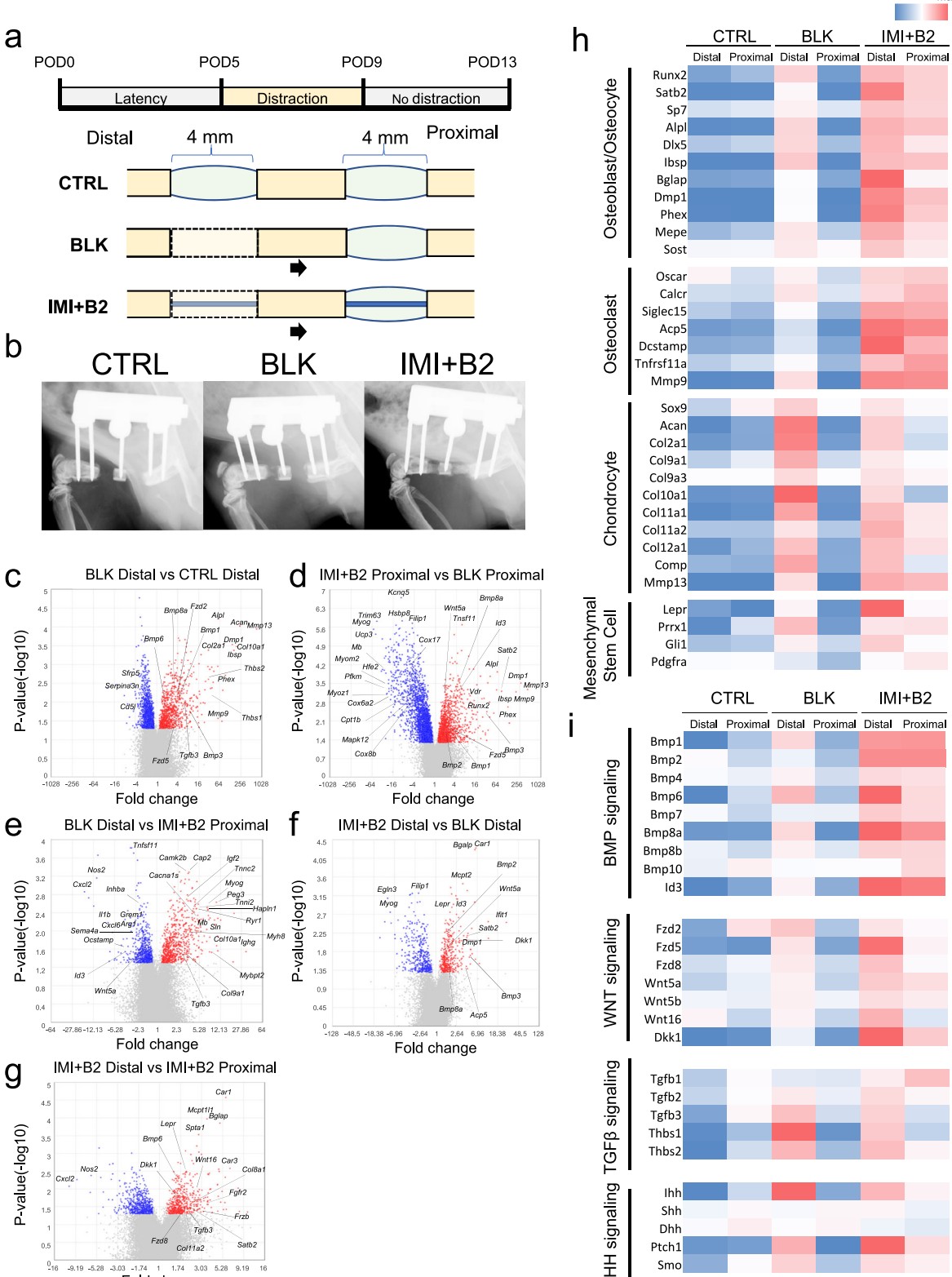

**Fig. 8 | The biological effect of DO and (IMI + B2) during the early stage of bone regeneration. a** Schematic illustration of the animal experiment. **b** Representative X-ray images of each group at POD13. **c**–**g** Volcano plots of differentially expressed genes by microarray analysis (*n* = 3 independent experiments per group). **h** Heatmap analysis of the cell type markers for osteoblasts/osteocytes,

chondrocytes, osteoclasts, and mesenchymal stem cells (MSC). The expression level of genes was quantified by microarray data; Blue, low expression; red, high expression. **i** Heatmap analysis of the gene expression related to BMP, WNT, TGF-β, and HH signaling. Source data are provided as a Source data file.

**HyTEC IM implant in a preclinical sheep bone transport model**

To determine the feasibility of the HyTEC IM implant technique in a large animal model, we established a preclinical sheep bone transport model using circumferential fixators as a proof-of-concept study. Following the creation of a 30-mm segmental defect in the sheep metatarsus (similar in size to the human tibia), a 3D printed, scaled-up IM implant (60 mm in length, 6 mm in diameter, 78% porosity, infiltrated with HyTEC gel) was placed in the intramedullary canal and secured by a small Kirschner wire through the distal end of the implant. Distraction over the IM implant was successfully performed using the same protocol as that applied in the rat model, indicating the translational relevance of this HyTEC IM implant technique (Supplementary Fig. 21).

## Discussion

In this study, we successfully established a novel long bone transport DO model in rat which mimics the clinical setting and outcomes of human patients subjected to bone transport surgery, which typically exhibits slow consolidation at the regenerate site and high morbidity rates of docking site nonunion and pin tract infection[14]. Furthermore, we engineered an osteoinductive and biodegradable IM implant and demonstrated its high efficacy to accelerate bone healing and address complications as an adjunctive therapy of bone transport in a single procedure.

BMP-2 and BMP-7 have been approved for clinical use in open fractures, nonunions and spinal fusion[37,38], but their delivery must be appropriate to achieve satisfactory clinical outcomes and avoid side effects. A delivery system for BMPs should retain the growth factors at the bone injury site for a prolonged period, providing optimal biodegradability and mechanical support for tissue ingrowth[39,40]. BMP-2 or BMP-7 has been locally applied by injection or collagen sponge to promote bone consolidation in bone lengthening DO in animal models[33,35,36,41], with dose-dependent effects[35]. Our study demonstrated that even a low dose (2 µg per rat) of BMP-2 incorporated into the IM implant effectively promoted bone regeneration in bone transport, enabling weight-bearing without secondary operations. This work presents a unique method of BMP-2 delivery through a biodegradable IM implant compatible with the DO procedure, minimizing interruption to the surgical procedures while promoting local bone healing.

In this study, osteoinductive IM implants were manufactured using a novel HyTEC technique by coating PCL-TCP filaments with a layer of BMP-2-laden hydrogel followed by freeze-drying. PCL-TCP composites are biocompatible, bioresorbable, osteoconductive, and clinically available[42–45], but lack osteoinductive factors[46]. Surface coating has been used to immobilize proteins on the surface of scaffolds for tissue engineering applications[47–49]. However, the amount of protein that can be loaded on the surface is typically low and the release rate is fast[47]. Hydrogels with their large water content and their porous microstructure provide a platform for adequate loading and sustained release of proteins[50–52]. However, loading soft hydrogels on the surface of rigid constructs is challenging due to mechanical mismatch at the interface. Furthermore, scaffolds loaded with protein-laden hydrogels via conventional techniques need to be used immediately to avoid hydrogel detachment and protein denaturation. IM implants in this work had a layer of BMP-2-laden hydrogel that was freeze-dried and integrated fully with the rigid filament surface via consecutive treatments to increase hydrophilicity, improve hydrogel adhesion, and stimulate surface-initiated crosslinking. NaOH treatment imparts hydrophilicity to the surface of polyesters. The pendant acryloyl moieties of GelMA onto the surface provide a covalent binding site with the GelMA and PEGDMA in the hydrogel. $CaSO_4$ microparticles were deposited and entrapped on the soft surface of the GelMA-modified filaments to facilitate the physical crosslinking of alginate[53]. The GelMA and PEGDMA macromonomers within the physically crosslinked hydrogel were covalently crosslinked in the next step to form a stiff interpenetrating network. HepMA was applied for its ability to act as a growth factor delivery vehicle through reversible binding with positively charged proteins, thereby facilitating sustained delivery and preservation of protein bioactivity[54]. In summary, the freeze-dried or rehydrated hydrogel coating consists of a robust interpenetrating network of physically crosslinked alginate and covalently crosslinked GelMA/PEGDMA, which exhibits strong adhesion to the surface of PCL-TCP. The adhesion is achieved through a combination of covalent binding of the hydrogel to the functional groups on the PCL-TCP surface and mechanical interlocking between the hydrogel macromonomers and cracked surface of the treated PCL-TCP. This robust adhesion ensures that the dried or rehydrated hydrogel coating remains firmly attached during and after implantation. As a result, this BMP-2-laden IM implant allows for interlocking fixation using the fixative pins and retention of an intact and stable hydrogel coating during bone transport. In the present study, BMP-2 could be released in a sustained manner for 21 days. However, a prolonged release of BMP-2 has been considered when it is applied to heal a large segmental bone defect in large animals or humans. In our study, additional coatings with PCL onto the IM implant significantly prolonged the release of BMP-2, which will be tested in the preclinical sheep model. Although we did not measure the degradation of the overall IM implant, previous studies suggest that such PCL-based scaffolds degrade in 2 years in humans[55], and the hydrogel-based coating layer consisting of PEGDMA, GelMA, and alginate is expected to degrade within a few months[56–58]. Therefore, the method presented here could be used for manufacturing storable, osteoinductive, and biodegradable IM implants to stimulate in vivo bone regeneration.

Docking site nonunion is a commonly encountered problem during bone transport, which leads to significant delays in the healing process, necessitating secondary bone grafting surgeries, and prolongation of fixator removal[59]. Docking site nonunion is typically attributed to the inactive bone contact and substantial soft tissue interposition[15]. As the intercalary segment approaches and closes the gap during the bone transport, the hematoma in the gap is gradually replaced with fibrocartilaginous tissue[15]. The leading edge of the intercalary segment and the docking site are also sealed over by the fibrocartilaginous tissue. Our bone transport models (controls) had a very low union rate at the docking sites, which is consistent with previous clinical outcomes[14]. Furthermore, our findings suggest that the IM implant itself does not affect the bone transport procedure.

Strategies to improve docking site fusion have traditionally relied on surgical interventions including acute shortening, bone grafting, compression, alternate compression-distraction, or bone marrow grafting combined with demineralized bone matrix[15,59]. Exciting results from this study showed that docking site union was achieved in all the animals treated with either 2 µg or 6 µg BMP-2 incorporated IM implants as early as POD34, with significant improvement in mechanical properties at both time points without any secondary operations or grafting. An improvement in mineralization was only observed during the early phase, but not during the late phase, which can be attributed to the release kinetics of BMP-2. The early phase data showed improved mineralization due to the sustained release of BMP-2 observed up to 21 days in our in vitro study. However, the effect was not sustained as all the BMP-2 was released, resulting in a decrease in mineralization after the initial 3 weeks of consolidation. Transcriptome analysis on POD13 suggests that BMP-2 eluting from IM implant enhances bony union at docking sites by promoting osteoblast/osteocyte differentiation and preventing skeletal muscle cells differentiation and growth, resulting in early bony fusion. In addition, an additive effect of eluting BMP-2 and DO on accelerating bone consolidation was observed early at the regenerate sites, as evidenced by higher expression levels of osteoblast/osteocyte and osteoclast markers via BMP and Wnt signaling. The sustained release of BMP-2 from

the IM implants was found to have an anabolic effect at the leading edge of the intercalary segment and the docking site by promoting active bone formation and facilitating bony fusion upon meeting. Despite the potential for IM implants to compromise bone marrow by occupying the medullary canal, clinical studies have demonstrated that metallic IM nails can provide mechanical support without delaying bone healing during bone transport procedures[17,19,23].

Bone consolidation at regenerate sites is another critical step for bone healing following distraction during the bone transport procedure. Although an additive effect of eluting BMP-2 and DO was observed at the very early phase (POD13), there was no significant change in bone mass among the groups at the regenerate site on POD34. This indicates that BMP-2 elution does not further enhance the DO-mediated bone regeneration after POD13. Notably, our BMP-2 eluting IM implants did not cause premature consolidation that may interfere with further lengthening or distraction procedures[35]. This suggests that the implants with appropriate dosage and release kinetics can accelerate consolidation in a controlled manner, thereby avoiding premature consolidation and holding promise for treating large bone defects.

Pin tract infection is one of the common complications of external fixation, with reported infection rates up to 100%[60]. Numerous factors can contribute to its development, including post-operation care, use of prophylactic antibiotics, pin design characteristics, surgical technique, and patient-specific risk factors[61]. Excessive activity after surgery can increase pin irritation and infection, particularly in uncontrollable animal models. In our study, pin tract infections were observed in the BLK, (CS + B2), IMI, and (IMI + B0.5) groups, with identified bacteria such as *Proteus* of fecal origin, indicating bacterial invasion to the poorly osseointegrated pins. Although those bacteria such as *Proteus* may not be clinically relevant in pin tract infections, animal grooming can introduce fecal origin to the pin tract sites, particularly in the case of loosening pins. Interestingly, there was no pin tract infection or pin loosening in the (IMI + B2) and (IMI + B6) groups, suggesting that BMP-2 released from the implants enhanced the osseointegration between pins and surrounding bone, and more importantly, accelerated bone healing in both docking and regenerate sites, leading to improved mechanical stability, reduced motion at the pin-bone interface, and lowered the loosening risk that can contribute to infection. Further research is needed to further decipher its anti-infection effect.

Several limitations exist regarding a broader extrapolation of the research findings from this current study. First, we did not include a (IMI + B6) group on POD55, as we aimed to achieve a satisfactory bone regeneration outcome while avoiding potential side effects by using a minimal dose of BMP-2. Second, the observation time points of 3 weeks or 6 weeks after consolidation in animal studies may not be long enough to evaluate the long-term effect of the implants in vivo. Future studies using longer time points and larger sample sizes are needed to assess the full load-bearing efficiency and implant degradation after fixator removal. Third, the use of a rodent model limited our IM implant design geometry to a rod-like structure, which has a relatively shorter growth factor release duration for the IM implant. Considering the broad muscles surrounding the femur in rodents, a unilateral external fixator rather than a circumferential one was applied to facilitate the operation. Future IM implants designed for large animal or clinical trials will need to be scaled-up with a prolonged release, which can be tested using a more clinically relevant circumferential fixator, as shown in our pilot sheep experiment[62,63].

## Methods

### Study design

The grafts were prepared as osteoinductive intramedullary implants to guide the regeneration of 8-mm femoral segmental defects over bone transport. A hybrid tissue engineering construct (HyTEC) strategy was developed. The implants comprise a composite of PCL-TCP (4/1, w/w) filaments coated with freeze-dried hydrogel loaded with BMP-2 (0.5, 2 or 6 µg) or without BMP-2 (Fig. 1a). The grafts were first characterized for their surface morphology ($n = 3$), tensile strength ($n = 7$), adhesion strength ($n = 6$), and protein release kinetics ($n = 3$) in vitro. The grafts were then implanted into the medullary canal of bone transport models, with 8-mm segmental defects surgically created in rat femurs (Fig. 2a). The bone transport models were chosen to vigorously evaluate the in vivo efficacy of the IM implants loaded with different doses of BMP-2 (0, 0.5, 2, or 6 µg) in providing sustained release of osteoinductive BMP-2 and functional bone regeneration in extreme long bone defects ($n = 8$). The rats subjected to bone transport only were regarded as surgical controls. The rats treated with bone transport and a collagen sponge (CS) loaded with BMP-2 (2 µg) at bone defect site were regarded as clinically relevant controls. The bone transport protocol of this study consisted of 5-day latency, 8-day distraction, and 21-day or 42-day consolidation (Fig. 2a). µCT analysis, mechanical testing, histology, histomorphometry, and pathogen identification were conducted to evaluate the efficiency of the IM implant in bone regeneration adjunctive to bone transport at the endpoint on post operation day 34 (POD34) or 55 (POD55). Skeletal mature (12 weeks) male Sprague Dawley (SD) rats were randomly assigned to different groups. Contralateral intact femurs were also collected and measured for normalizations in bone mass and mechanical properties. The early genetic responses of BMP-2 and DO at the regenerate sites and docking sites were measured by microarray analysis. Power analysis for the in vivo study ($n = 8$ per group per time point) is detailed in the "Statistical analysis" section. Sample size "$n$" refers to per treatment group per time point in this paper, except where otherwise indicated. Femoral samples ($n = 8$) were harvested from rats on POD34 (3 weeks of consolidation phase) or POD55 (6 weeks of consolidation phase) for µCT analysis ($n = 8$), mechanical test ($n = 8$), histology ($n = 5$), or histomorphometry ($n = 3$). The assessments above were carried out using the same samples sequentially in a randomized and blinded fashion.

### Chemicals

Medical-grade polycaprolactone (PCL, $M_n = 80$ kDa) was purchased from Sigma-Aldrich. β-TCP nano-powder with average particle size of 100 nm (TCP) was purchased from Berkeley Advanced Materials Inc. N,N-Dimethylformamide (DMF), sodium hydroxide (NaOH), and ethanol were purchased from Fisher Scientific Inc. N-(3-Dimethylaminopropyl)-N′-ethylcarbodiimide hydrochloride (EDC), N-Hydroxysulfosuccinimide (NHS), 2-(N-Morpholino)ethanesulfonic acid (MES), N-(3-Aminopropyl)methacrylamide hydrochloride (APMA), gelatin type A, and heparin, and Calcium sulfate dihydrate (CaSO$_4$) were purchased from Sigma-Aldrich. Polyethylene glycol dimethacrylate (PEGDMA, $M_n = 1000$ gr/mol) was received from Polyscience, Inc. Sodium alginate (alginate, 500 GM) was purchased from Pfaltz & Bauer Inc. Human BMP-2 protein was provided by Medtronic. Human BMP-2 ELISA kit was purchased from Sigma-Aldrich.

### Polymer synthesis

PCL-TCP (4/1, wt/wt) was synthesized and extruded into homogeneous filaments in a diameter of 1.0 mm and a length of 18.0 mm as described[64]. Briefly, 80 g of PCL and 20 g of TCP were separately dissolved in 800 mL and 400 mL of DMF, respectively and stirred for 3 h at 80 °C. The PCL and TCP solutions were then mixed, and the mixture was stirred for an hour. Then the mixture was precipitated in 4 liters of water to make PCL-TCP composite sheet. The PCL-TCP composite sheet was rinsed with water and residual solvent was evaporated inside of a fume hood at ambient temperature for 24 h. The dried PCL-TCP composite sheet was cut into pellets and extruded using a custom in-house filament extruder. To synthesize gelatin methacrylate (GelMA) macromonomer, gelatin was dissolved in deionized water (10% w/v) at

50 °C. Methacrylic anhydride was added to gelatin solution at a molar ratio of 100:1 (methacrylic anhydride:gelatin) and the solution could react under stirring for 1 h at 50 °C. The mixture was then 5× diluted with deionized water and dialyzed against deionized water using a dialysis tube (Spectrum Laboratories, Rancho Dominquez, CA) with 6−8 kDa molecular weight cutoff for 3 days at 40 °C. The GelMA solution was then freeze-dried and stored at −80 °C. To synthesize methacrylated heparin (HepMA), 1 g heparin was dissolved in 100 mL MES buffer (100 mM). 5 mL MES buffer containing 45 mg EDC and 30 mg NHS was then added to the heparin solution to activate the carboxylic acid. After 1 hr reaction at room temperature, 25 mg APMA in 1 mL MES was added to the solution and allowed to react for 2 hrs at room temperature. The HepMA solution was then dialyzed against deionized water using a dialysis tube (Spectrum Laboratories, Rancho Dominquez, CA) with 6−8 kDa molecular weight cutoff for 3 days at ambient temperature, lyophilized, and stored at −80 °C.

### Coating PCL-TCP filaments with BMP-2-laden hydrogel

PCL-TCP filaments were treated in consecutive steps to increase hydrophilicity, improve hydrogel adhesion, and stimulate surface-initiated crosslinking (Fig. 1a). The filaments were then coated with BMP-2-laden (0.5, 2, or 6 μg) or without BMP-2 (0 μg) hydrogel. PCL-TCP filaments with 1.0 mm in diameter were synthesized, manually cut to make 18 mm filaments, and dipped into a 5 N NaOH solution for 6 hrs. The filaments were then washed three times with deionized water and incubated in an MES buffer (100 mM) containing EDC (5 mg/mL) and NHS (5 mg/mL) for 30 mins at room temperature in order to activate the carboxylic acid groups on the surface. Then, the filaments were washed three times with deionized water and incubated in GelMA 2% solution in MES buffer for 1 hr at 37 °C. The filaments were then washed three times with deionized water to remove the unreacted GelMA and incubated in EDC/NHS (5 mg/mL) in MES buffer solution for 15 mins at room temperature. The GelMA coated filaments were then washed three times with deionized water and dried under vacuum. Then, the GelMA coated filaments were dipped into a CaSO$_4$ suspension in deionized water (100 mg/mL) at 60 °C and sonicated for 30 secs. The filaments were then transferred into wells of a 24-well plate and dried under vacuum. The dried filaments were dipped into wells of a 96-well plate containing GelMA (15%), Alginate (1.25%), PEGDMA (2%), HepMA (1%), protein (BMP-2, 200 μg/mL), and photo-initiator (0.3%) in deionized water at 37 °C for 2 mins. The hydrogel-loaded filaments were removed from the solution and left in dry wells of another 96-well plate for 5 mins. The hydrogel-loaded filaments were then irradiated with visible light for 15 mins to covalently crosslink GelMA, PEGDMA, and HepMA. The crosslinked hydrogel-loaded filaments were stored at -80 °C and freeze-dried. For the coating of additional layers of PCL, the hydrogel-loaded filaments were freeze-dried before quick immersion into a PCL solution (PCL dissolved in chloroform at -20 °C) for 2 s before returning to room temperature. BMP-2 (2 μg/sample) or FITC-conjugated BSA (200 μg/mL) was incorporated into the freeze-dried hydrogel before PCL coating. The BMP-2-laden HyTEC graft was coated with a single layer or triple layers of PCL and make modified HyTEC graft (mHyTEC).

### Tensile modulus, hydrogel loading, surface morphology and contact angle characterization

Tensile modulus of PCL-TCP filaments was measured using an Instron 5944 uniaxial testing system with a 2 kN load-cell (Instron Corporation, Norwood, MA) and 1 N pre-load. Tensile modulus measurement was performed at a displacement rate of 1% strain/s up to 25% strain. The slope of the linear regime of the stress vs strain curve was taken as the tensile modulus. 5 samples per group were used for tensile modulus measurements.

The hydrogel loading (%) was calculated from the scaffold weight before hydrogel loading (Wb) and after hydrogel loading (Wa), using the following equation. 8 samples per group were used for hydrogel loading measurements.

$$hydrogel\ loading = \frac{Wa - Wb}{Wb} \times 100$$

For visualizing the surface morphology, the filaments were immersed in liquid nitrogen and freeze-dried after treatment with NaOH, freezing/thawing, or hydrogel coating. To visualize the interface between filament and hydrogel, the hydrogel-coated filaments were freeze-dried and then cut cross-sectionally or longitudinally using a razor blade before mounting onto pin stub specimen holders using conductive paint (PELCO Conductive Silver Paint, Ted Pella, USA). The samples were coated with a thin layer of gold and palladium (Au/Pd 60:40) (Cressington 108 Auto Sputter Coater, Cressington Scientific Instruments, Watford, United Kingdom) and imaged using a low vacuum scanning electron microscope (Apreo S LoVac, Thermo Fisher Scientific) at an accelerating voltage of 3 keV.

Contact angle goniometry at the filament-hydrogel interface was conducted on the longitudinal samples by a Contact Angle Goniometer (Rame-Hart 290, Rame-Hart Instrument Co, NJ, USA). Water droplets (2 μL) were applied at the center of the freeze-dried filament-hydrogel interface, repeated 10 times, and averaged.

### Interface shear strength

A customized 3D printed PCL-TCP device was designed and used to evaluate the adhesion of the HyTEC hydrogel to PCL-TCP scaffolds. The device was composed of two concentric cylinders separated with a 1.5 mm gap filled with hydrogel and connected through two bridges. The hydrogel was made within the gap between two concentric cylinders and then freeze-dried for 2 days. The freeze-dried hydrogel-incorporated device was then placed on an Instron 5944 uniaxial testing system (Instron Corporation, Norwood, MA) with a 100 N load-cell and the bridges connecting inner and outer cylinders were cut. The test was performed by applying force on the inner cylinder using a 0.1 N preload and 0.1 mm/s displacement rate. The interfacial shear strength represents the peak shear stress at the interface of the freeze-dried gel and the PCL-TCP part. The experiment is conducted on 3 samples ($n = 3$) and the resulting interfacial stiffness was calculated based on the stain-stress curves.

### Protein release

For measurement of release kinetics, hydrogel-coated filaments with 0.5, 2 or 6 μg encapsulated BMP-2 were freeze-dried and incubated in 1 mL PBS at 37 °C for 28 days. At each time point, the amount of BMP-2 in the release medium was measured using ELISA and the release medium was replaced with fresh PBS. In order to investigate the effect of implant storage on the protein activity and release kinetics, BMP-2-laden hydrogel-coated filaments were stored for 2 months at 4 °C and then the release kinetics of BMP-2 from the stored implants was measured and compared with those of freshly made implants. In addition, the release kinetics of FITC- BSA or BMP-2 were further measured weekly within 91 days (13 weeks) or 28 days (0, 1, 3, 7, 14, 21 or 28 days) directly by a plate reader or by ELISA, respectively.

### External fixation device design for bone transport

The customized external fixator device consists of two parts: one frame (32.0 mm in length) and five fixative pins (1.2 mm in diameter, 22.0 or 17.0 mm in length) with threads (1.0 mm in diameter, 0.5 mm in length). The frame has two fixed ends, which were used to lock two pins at each end, and to fix proximal and distal bone segments after osteotomy. The frame also has one movable part between the two fixed ends, which was used to fix an intercalary segment after corticotomy for bone transport. The performance of the devices was tested

ex vivo in a rat femoral specimen by transporting bone segment in a retrograde direction.

## Surgical procedures

The animal experimental protocol was approved by the Institutional Animal Care and Use Committee (IACUC, 33395) of Stanford University following ARRIVE guidelines. A novel external fixation device was customized for bone transport in rat. 12-week-old male SD rats weighing 350 to 400 g were purchased from Charles River Laboratories. All surgeries were performed under anesthesia by 3 to 5% isoflurane-oxygen (VetOne, Boise, ID). Buprenorphine Sustained-Release (1 mg/kg; ZooPharm, Laramie, WY) and Carprofen (5 mg/kg; Zoetis, Parsippany, NJ) were administered subcutaneously to minimize the suffering of the animals before surgery and post-op for 3 days, respectively. Cefazolin (25 mg/kg; TCI, Tokyo, Japan) was also given subcutaneously post-op for 3 days. 8-mm femoral segment defect was surgically created and stabilized with the external fixator. The surgical procedures were modified from a previous study[65]. Briefly, upon incision, the femur was exposed from surrounding muscles. Five fixative pins were drilled vertically into the lateral side of femur using a drill guide. An 8-mm transverse osteotomy was created at the midshaft by a wire diamond saw (RISystem, Landquart, Switzerland). Another 4-mm transverse corticotomy was performed for bone transport. The prepared grafts with a length of 18.0 mm and diameter of 1.2 mm were inserted into the medullary canal of proximal and distal femoral segments, with the two ends secured by the two fixative pins close to the defect in a press-fit way. The IM implants were coated with 0.5 μg (IMI + B0.5, $n = 8$), 2 μg (IMI + B2, $n = 8$) or 6 μg (IMI + B6, $n = 8$) BMP-2-laden hydrogel as described above. The rats subjected to bone transport only were regarded as surgical controls (BLK, $n = 8$). The rats treated with bone transport as well as implantation of collagen sponge (CS, Integra LifeSciences) cylinders (3 mm in diameter, and 8 mm in length) preabsorbed with BMP-2 (2 μg) were put into the bone defect and regarded as clinically relevant controls. Surgical incisions were then sutured sequentially. The bone transport protocol of this study consisted of three phases: a latency phase of 5 days, an 8-day active lengthening phase (0.5 mm/12 h), and a consolidation phase of 21 (POD34) or 42 (POD55) days (Fig. 2b). Bone transport was performed over the IM implant in a retrograde direction (Fig. 2a). The grouping methods and related assessments are summarized in Table S1.

## Post-op care and sample harvest

Animals were housed individually after operation. Pin tract infection was monitored and managed in all the animals during the study period. Polyvinylpyrrolidone iodine and ethanol were used during pin tract care when required. Three rats ($n = 3$ per group per time point) were randomly selected and received subcutaneous injection of xylenol orange (30 mg/kg, Sigma-Aldrich, St. Louis, MO) at 13 days before termination and calcein (10 mg/kg; Sigma-Aldrich) at 3 days before termination for in vivo labeling. At the endpoints (POD34 or POD55), the fixative pins and frames were removed from femurs, and the femurs were harvested after euthanasia with carbon dioxide. To observe weight bearing efficacy, fixators were removed in one of the rats in IMI and (IMI + B2) group 1 week before POD55. Upon harvested, the femoral specimens were initially fixed in 10% formalin for 48 h, then transferred to 70% ethanol for further assessments.

## Micro-computed tomography (CT) analysis

Microstructural change within the distraction regenerate site and docking site was qualitatively and quantitatively assessed using μCT. Briefly, all the specimens were scanned by Skyscan 1276 μCT (Bruker, Kontich, Belgium) at a custom isotropic resolution of 20 μm isometric voxel size with a voltage of 70 kV and a current of 200 μA. Post processing of the reconstructed images was analyzed using the software packages including SkyScan NRcon, CTAn, and CTVox (Bruker, Kontich, Belgium). Two regions of interest (ROIs) were analyzed separately, including the regenerate site (8 mm in length, 400 slices) and the docking site (2 mm in length, 100 slices). A global threshold of 80 (minimum bone densities of 380 mg HA/cm$^3$ and above) was applied to remove graft and soft tissue background. Cross-sectional slices of the lengthening zone or docking site were used for bone tissue volume fraction (bone volume/total volume, BV/TV) measurement by CTAn. BV/TVs of the ROIs were normalized by those in contralateral intact control. 3D bone structure was made from the segmented dataset with CTAn (CT Hounsfield units (HU) threshold > 10,000) for visual inspection using the MicroView 3D Image Viewer (Parallax Innovations Inc., Ilderton, Canada). Nonunion was confirmed by μCT analysis, which was defined by no radiographic evidence of bone fusion at the regenerate site or docking site.

Osseointegration of pins were also determined by micro-CT analysis. The data for analysis were extracted from those in the whole femoral samples scanned by Skyscan 1276 micro-CT (Bruker, Kontich, Belgium). The scanning methods were shown in the manuscript. The evaluation was done along the pin thread, resulting in about 50–100 slices per sample. A circle with a diameter 0.8 mm bigger than the thread of the pins (1.0 mm in diameter) was chosen as the region of interest for evaluation of the bone volume per total tissue volume (BV/TV) (Supplementary Fig. 16).

## Mechanical test

Mechanical properties of specimens were evaluated by three-point bending tests. A material testing system (Instron 5944 testing system, Norwood, MA) with a 2 kN load cell was used to test the femurs to failure. The femurs were loaded in the anterior-posterior direction with a span of the supportive blades set as 20 mm. The bones were tested at a speed of 0.05 mm/s, with the long axis of the femur placed perpendicular to the blades during the test. The modulus of elasticity in tension (Young's modulus), maximum load, and energy to failure were obtained and analyzed with built-in software (Bluehill Universal; Instron, Norwood, MA). The biomechanical properties of the new bone were expressed as percentages of the contralateral intact bone properties. During the mechanical tests, we ended the compression testing once the loading showed a 15% decrease to prevent breaking of the bone.

## Microarray analysis

To measure the early response of BMP-2 and DO in bone transport, tissues extraction from regenerate sites (distal) and docking sites (proximal) were subjected to microarray analysis on POD13. In this experiment, 8 mm bone defect and 4 mm segmental bone were created in rat left femurs. The rats were subjected to bone transport for 4 days without an implant (BLK, $n = 3$) or with a HyTEC IM implant incorporated with 2 μg BMP-2 (IMI + B2, $n = 3$). Other rats with acute bone transport to the same location (4 mm apart from distal and proximal ends) on the date of osteotomy were regarded as the control (CTRL, $n = 3$). For rats in the CTRL group, segmental bone was placed in the middle of the defect site at the day of surgery, and segmental bone distraction was not conducted. Rats in the BLK group received segmental bone distraction (0.5 mm/12 hr) from POD 5 to POD9. Rats in the (IMI + B2) group were implanted (IMI + B2) at the day of surgery, and segmental bone was distracted at the same rate as the BLK group. All rats were euthanized by $CO_2$ at POD13, and X-ray (UltraFocusDXA; Faxitron Bioptics, Tuscon, AZ) was taken before collecting the tissue samples. Tissue samples at each defect site were collected and were immediately frozen in liquid nitrogen ($n = 3$).

To extract total RNA, tissue samples were disrupted and homogenized in Qiazol Lysis Reagent (Qiagen, Germantown, MD) using a Bullet blender tissue homogenizer (Next Advance Inc., Troy, NY, USA). Then, samples were processed using a RNeasy universal mini kit (Qiagen, Germantown, MD). The quality of RNA was assessed using the

RNA 6000 Nano Kit (Agilent Technologies, Santa Clara, CA). Micro-array was performed with the Rat Clariom D assay (Affymetrix, Santa Clara, CA). Samples were fragmented and labeled with the GeneChip WT Terminal Labeling and Controls kit and hybridized to the arrays with the GeneChip Hybridization Wash and Stain Kit according to the manufacture's protocol (Affymetrix). The arrays were washed and stained on the GeneChip Fluidics Station 450. The arrays were scanned with the GeneChip Scanner 3000 7 G. Analysis was performed using Transcriptome Analysis Console (TAC, Affymetrix) 4.0 software. Data were normalized with MAS 5.0 algorithm (Affymetrix). Volcano plot figures were created in TAC. Heatmaps were created in Microsoft Excel. All genes with adjusted $p$-value < 0.05 and fold change > 1.2 were used for the analysis.

## Histology and immunohistochemistry

The femoral specimens ($n = 5$ per group) were decalcified in 10% EDTA solution for 5 weeks and embedded in paraffin after dehydration with ethanol. Thin sections (5 μm) were cut by a microtome (RM2525, Leica, Germany) along the long axis of each femur in the sagittal plane or cross-sectional axis of pin tracts at the two ends of femurs. The slides were stained with hematoxylin and eosin (H&E; Sigma-Aldrich, St Louis, MA), Masson Trichrome staining (Abcam, Cambridge, UK), or Tartrate-resistant acid phosphatase (TRAP; Sigma-Aldrich, St Louis, MA). Immunohistochemistry was conducted by incubating the anti-bodies including anti-osteocalcin in 1:100 (sc-365797; Santa Cruz Bio-Tech, Dallas, TX), anti-CD31 in 1:100 (sc-376764; Santa Cruz BioTech, Dallas, TX), anti-BMP-2 in 1:200 (ab6285; Abcam, Cambridge, UK), or anti-VEGF antibody in 1:100 (ab1316; Abcam, Cambridge, UK). A horseradish peroxidase-streptavidin detection system (Dako, Santa Clara, CA) was used, followed by counterstaining with hematoxylin.

## Histomorphometry

For histomorphometry, a protocol for paraffin embedding of miner-alized bone was applied in this study as described previously[66]. After fixation, the specimens ($n = 3$) were treated with 5.0% (w/v) aqueous potassium hydroxide for 96 h at room temperature on an orbital shaker. Then the bones were washed with water and then dehydrated in ethanol under a vacuum infiltrating system. Processed bones are embedded routinely into paraffin blocks. 10-μm sections were cut by a RM2255 microtome (Leica, Wetzlar, Germany) along the long axis of each femur in the sagittal plane. For histomorphometry, two sections with 100 μm apart were selected for measurements. Fluorescent ima-ges were taken under an All-in-One Fluorescence Microscope BZ-X800 (Keyence, Osaka, Japan). 3 or 6 random images at 10× magnification at docking site or regenerate site of each sample were applied for mea-surements. Mineral apposition rate (MAR) was determined by the distance between in vivo labels, divided by the 10 days interval.

## Classification of pin tract infections by visual inspection

On POD34 or POD55, pin tract infections were evaluated using a modified Checketts classification before sacrificing the animals[67]. Grade 0 corresponds to "no redness", in which no redness, discharge, or pin loosening was observed. Grade 1 corresponds to infections only in the soft tissue, characterized by redness and discharge around the pin without pin loosening. Grade 2 corresponds to infections in both soft and bone tissues, characterized by redness and discharge around the pin associated with pin loosening caused by osteomyelitis. A single physician who was blinded to the group identification of pins eval-uated the rats for pin tract infections. The animals presented infection from Grade 1 to Grade 2 in one pin site at least were regarded as infected. Other animals were regarded as non-infected.

## Bacterial identification

Bacteria were isolated directly from all fixative pins after removal in IMI or (IMI + B2) groups on POD34 or POD55 (2 rats with 10 pin sites in total per time point), respectively (Table S2). The pins were screwed out from each animal ($n = 2$) by sterile instruments after euthanasia. The pins were then washed twice in sterile PBS and then sonicated at room temperature for 30 min to remove the biofilm in the TB medium. The TB medium was then transferred to a 37 °C shaker and incubated for 16 h. The smear formed in the TB medium was observed. The colony formation was examined after the TB medium (30 μl) added onto the agar gel plates and incubated for another 16 h in 37 °C incubator. The colony formation positive area was determined by the colony area/total area in the plate (10 plates per group per time point).

Bacterial genomic DNA was isolated from 2 ml of TB medium from each sample using a QIAamp DNA kit (Qiagen, Hilden, Germany) and quantified using a Nanodrop Spectrophotometers (Thermo Fisher, Waltham, MA). All extracted DNA samples were stored at −20 °C until further analysis. High-throughput, next-generation sequencing (NGS) based on 16 S rDNA gene amplicons was used to identify the bacteria. NGS library preparations and Illumina MiSeq sequencing were con-ducted by GENEWIZ, Inc. (South Plainfield, NJ). For the library pre-paration, a library sequence of the V3 and V4 hypervariable regions of 16 S rDNA was constructed using a 20-ng DNA aliquot isolated from each sample. The 16S-EZ workflow entails proprietary multiplexed PCR amplification of the V3 and V4 hypervariable regions of the 16 S rDNA gene. A limited cycle 2nd round PCR adds sample-specific barcodes to each sample to allow for multiplexing multiple samples together on the same sequencing run. Final libraries are pooled together and undergo final quality control (QC) prior to sequencing. Sequencing is performed with Illumina chemistry in the 2 × 250 bp paired-end con-figuration to ensure the entire V3 and V4 sequences are covered. Subsequent sequencing data undergoes a rigorous QC prior to per-forming 16S-EZ bioinformatics analysis. The sequencing data quality was optimized before Operational Taxonomic Unit (OTU) analysis and species annotation. In order to obtain the classification information of OTU, a representative sequence was selected for each OTU and annotated using the Ribosomal Database Project (RDP) classifier, thereby to obtain the community composition of each sample using Quantitative Insights into Microbial Ecology (Qime, version 1.9.1) software. For each sample, the percentage of each species at different taxonomic levels (Phylum, Class, Order, Families, Genus, Species) was determined afterwards.

## Preclinical study using a sheep bone transport model

This model was modified based on the Lopez-Plioeho's study, with a clinically relevant circumferential fixator[68]. Two skeletally mature male Rambouillet Cross ewes aged three to 4 years old with a mean weight of 71 kg and a metatarsal length >14 cm were selected. Both animals were deemed free of disease or condition that would interfere with the study and had never been used for previous testing. Approval for the surgeries was granted by the Colorado State Institutional Animal Care and Use Committee (approved protocol KP 1579). ARRIVE guidelines for reporting animal research were followed. Animals were monitored daily by veterinarians and qualified personnel for general health, neurologic problems, and lameness.

Under anesthesia, an optimized circular external fixator (IMEX, Longview, TX) was implanted on the right hind metatarsus. The frame was positioned and then fixed by six stainless steel centrally-threaded (4.0 mm shaft diameter, 4.8 mm thread diameter) full-pins. Each hole was predrilled with a 3.9 mm drill bit, using a guide. Three pins were placed in the distal portion of the metatarsus and three pins were placed in the proximal portion of the metatarsus, leaving at least 0.5 cm of bone between the osteotomy sites and the closest proximity pins. Two osteotomies were then performed with an oscillating saw to create a 3 cm defect with removed periosteum. The defect size in metatarsus of sheep is similar to the human tibia[69]. Next, three uni-cortical half pins (3.2 mm shaft and thread diameter, 3.1 mm predrill diameter) were implanted to fix the transportable bony segment to the

central ring, and a Gigli saw was used to release the 2 cm bony segment. A 3D printed, scaled-up IM implant of 60 mm in length and 6 mm in diameter, 78% porosity infiltrated with HyTEC gel without BMP-2 was inserted retrograde through the ostectomy site and placed mid-diaphyseal within the medullary canal of the metatarsus through the transportable bone. A 0.6 mm diameter Kirschner wire was drilled distal to the bony segment and through the IM implant to fix it in place during bone transport. The bone transport protocol of this study consisted of three phases: a latency phase of 5 days, a 30-day active lengthening phase (0.5 mm/12 h), and a consolidation phase of 6 months. X-ray images were taken monthly after implantation. The ewes were allowed to ambulate immediately after the surgery. Post-operative pain was managed by administration of Carprofen (4 mg/kg, s.c.) and transcutaneous Fentanyl patches. The surgical sites were bandaged to maintain cleanliness of pin tracts. The pin tracts were monitored and cleaned continuously throughout the study period. At the completion of the study period, the ewes were humanely euthanized.

## Statistical analysis

A priori power analysis (G*Power, Universität Düsseldorf) based on the previous studies using a similar rat bone lengthening DO model in evaluating a bone graft or stem cell performances determined that a sample size of 8 was needed to obtain 90% statistic power at the significance value of 0.05 when comparing quantitative μCT analyses outcome[65]. All statistical analyses were performed using Prism 7.0 (GraphPad Software Inc.). Shapiro-Wilk normality testing was used to evaluate data distribution. Pair-wise comparisons passing normality test were analyzed with Student's $t$-test while the Mann-Whitney rank sum test was used for pairwise comparison of nonparametric data. Multiple comparisons passing normality test were analyzed with ANOVA and post hoc Turkey's multiple comparisons test, whereas nonparametric multiple group comparisons were analyzed using the Kruskal–Wallis test with Dunn's post hoc testing. For μCT analysis and mechanical testing, contralateral femurs were used to normalize the parameters. Data of mechanical testing were presented as scatter dot plot with median and interquartile range (nonparametric). All the other quantitative data were presented as mean and standard error of mean (SEM), and the level of significance was set at $p < 0.05$.

## Reporting summary

Further information on research design is available in the Nature Portfolio Reporting Summary linked to this article.

# Data availability

The authors declare that all data supporting the findings of this study are available within this paper and its Supplementary Files. The microarray data are available in the GEO database under the accession numbers GEO: GSE200518. The bacterial 16 S rDNA sequencing data are available in the SRA database under the accession number: PRJNA983725. Source data are provided with this paper.

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

## Acknowledgements

We thank Dr. Ning Zhang, Dr. Youngbum Park, Dr. Alexander Stahl, and Dr. Liming Zhao for their technical support. We also thank Stanford Center for Innovation in In vivo Imaging (SCi3) small animal imaging center for providing imaging facilities for this project. This project is also supported by the funding bodies: National Institutes of Health grant R01AR057837 (Y.P.Y.), National Institutes of Health grant U01AR069395 (Y.P.Y.), National Institutes of Health grant R01AR072613 (Y.P.Y.), National Institutes of Health grant R01AR074458 (Y.P.Y.), Department of Defense grant W81XWH-22-1-0189 (Y.P.Y.), and National Institutes of Health grant 1S10OD02349701 (Timothy C. Doyle). Part of this work was performed at the Stanford Nano Shared Facilities (SNSF), supported by the National Science Foundation under award ECCS-2026822.

## Author contributions

S.L., H.M., S.M., and Y.P.Y. conceived and designed the study. S.M. and E.L. conducted polymer synthesis, graft fabrication and characterization. S.L., H.M., and S.M. performed animal surgeries and post-op care. S.L., H.M., S.M., E.L., H.V.A., J.L. and S.K. performed imaging, mechanical tests and histology. S.L. performed bacterial culture and identification. M.P., B.C.G., J.T.E., J.Y., M.G., D.M., W.J.M. designed and performed the sheep study. S.L., H.M., S.M., E.L., H.V.A., J.L., and S.K. and Y.P.Y. analyzed and interpreted the data. Y.P.Y. acquired funding and supervised. S.L., H.M., and S.M. wrote the original draft and E.L. and Y.P.Y. reviewed and edited.

## Competing interests

The authors declare no competing interests.
