## [Peer Review File · Nature Communications]

REVIEWER COMMENTS

Reviewer #1 (Remarks to the Author):

The purpose of this study is to propose a novel strategy to heal an 8mm bone defect in a rodent model using an osteoinductive biodegradable intramedullary implant loaded with BMP-2. There are a number of major issues of concern associated with this manuscript.

1. The authors propose combining an external fixator with an intramedullary implant. I have great concerns regarding the clinical relevance of this animal model. To heal large bone defects in humans it will be difficult to use a unilateral external fixator without having significant pin tract problems. This fixator may need to be in place for months. This is why circumferential thin wire fixators are used.
2. The IM implant will deliver the BMP-2 but at what rate will it biodegrade? What will be the impact of the implant on the surrounding tissue and bone formation? The implant is a potential source of infection.
3. The authors chose CT scan results as the outcome measure. In most bone healing studies of this nature biomechanical testing is used as the outcome measure. It is important to determine the quality of the bone that is present at both the distal docking site and the proximal regenerate. Furthermore, the study appears to be underpowered.
4. It is not clear from the data presented how many bone defects actually healed on plain radiographs or even CT scans with 3D reconstruction.
5. It appears that BMP-2 is secreted for approximately 21 days. This may not be a sufficient duration of production in trying to heal large segmental defect in humans.
6. In a human, how would the authors stabilize the intramedullary implant? If there is micromotion of the implant this could impact bone healing.
7. A Table delineating each group and the various tests done, etc. would be helpful to the reader.

Reviewer #2 (Remarks to the Author):

The manuscript by Yang and colleagues under consideration at Nature Communications describes a tissue engineering approach for sustained delivery of BMP-2 for bone regeneration. This IM implant model and other procedures would benefit from controlled release of bone repair morphogens. This is a carefully designed translational study with considerable evidence for successful procedure with enhanced osseointegration. The following points should be taken into consideration in a revised manuscript:

1) The rationale for the complex coating procedure is not well laid out. Why would you need GelMa then alginate followed by GelMA and PEGDMA after freeze drying? Some description of the role for mediating delivery, coating stability, etc. is needed in the results and discussion.

2) The data indicating dynamic mineralisation at day POD34 is compelling for the IMI+Bx groups. However, it is not clear why there is no significant difference at POD55. This should be discussed.

3) Evidence for increased osteogenesis and decreased rate of infection is clear. However, one of the potential caveats of integrating a soft/composite layer at the interface of a hard biomaterial is the potential for a weak interface. Some evidence that the new coating has at least comparable long term adhesion to the bone is important.

4) Another complication could be shelled particles that might trigger inflammation and potentially bone resorption. Considering the increase in osteoclasts, I am wondering if this could lead to bone resorption around the structure at later time points.

5) Page 8 Line 17: "Coating of the surface with GelMA presented double bonds on the surface for covalent linking to the hydrogel." This statement is confusing. Are the authors referring to pendant acryloyl moieties for further modification? This needs to be reworded with the surface chemistry explicitly defined. If the authors have performed rigorous characterisation of the materials interface in previous work, this should be discussed in more detail. Alternatively, more details of the interfaces between the multiple materials is important for the broad readership.

6) HepMA incorporation is described in the discussion but not in the results. More details are required for describing and evaluating the soft materials that facilitate sustained BMP release.

Minor:

Page 4 Line 13 – Extend should read Extent

Page 4 line 24 –compared should read comparison

Point-by-point response to the reviewers' comments

Reviewer #1 (Remarks to the Author):

The purpose of this study is to propose a novel strategy to heal an 8mm bone defect in a rodent model using an osteoinductive biodegradable intramedullary implant loaded with BMP-2. There are a number of major issues of concern associated with this manuscript.

1. The authors propose combining an external fixator with an intramedullary implant. I have great concerns regarding the clinical relevance of this animal model. To heal large bone defects in humans it will be difficult to use a unilateral external fixator without having significant pin tract problems. This fixator may need to be in place for months. This is why circumferential thin wire fixators are used.

Response:

We agree with the comments from the reviewer. Although there is a well defined technique and device manufactured by multiple companies using a monolateral transport frame for bone defect treatment (1), we understand that circumferential fixators are more routinely used in bone transport. It is also very important to establish a clinical relevant animal model in this study. In this study, we had considered designing a bilateral external fixator for bone transport for rat. However, there are some challenges we have faced. First, the local anatomic condition of rat femur does not allow a bilateral or circular fixator. The broad muscles surrounding rat femur favors a unilateral external fixator. Second, a unilateral fixator means a lower bearing weight for rat, which reduced disruption to daily movement of the animal.

Although there is a commercially available fixator for bone transport in mouse produced by RIS system which is also an unilateral external fixator, there is still no such fixator for rats. We then custom-made our fixator based on a bone lengthener to facilitate the intramedullary implantation in rat, which is technically difficult to do in mouse. We believe that a mechanically stable and reliable fixation is very important to get reproducible study outcomes. Before the animal experiments, we have done some pilot studies using the fixator samples first to improve the stability, feasibility of implantation, and reduce interruption to animals, with the help from our collaborative company which is professional in making fixators for patients. We have addressed this point as a limitation of rat model in the discussion (page 10, line 39-43).

However, we understand that this kind of unilateral fixator is not an ideal one for large animals or patients. As such, we performed additional preclinical sheep experiments to address this important clinical relevance concern. In our pilot sheep experiment, we established a preclinical sheep bone transport model using circumferential fixators based on literature (2,3) and demonstrated the feasibility of surgical handling, implantation and distraction of our scaled-up IM implant device. In the Lopez-Plioho's study (2), a 15-mm bone defect and 25-mm transportable bone segment were surgically created in the sheep metatarsus, which is similarly sized to the human tibia, and the latency and distraction periods were 7 days and 14 days, respectively. This model was modified to test and validate the clinical relevance of our device. Briefly, we surgically created a 30-mm critical size segmental bone defect and a 20-mm transportable bone segment in the metatarsus as shown in **Figure R1**. The sheep metatarsus is greater than 14 cm, and similar in size to the human tibia. The 30-mm bone defect in sheep metatarsus represents a loss of approximately 25% of the mean

length, which Boyce et al. suggest equates to roughly a 10 cm defect in the human tibia (4). We were able to distract the transportable bone segment at 1mm/day in two steps (0.5mm every 12 hours) as in clinic. Based on the sheep metatarsal anatomy and the surgical bone transport model, we designed and 3D printed a scaled-up IM implant of 60 mm long and 6 mm in diameter, and were also able to place the IM implant into the marrow cavity and placed a tiny k-wire (0.6mm in diameter) through the implant on the distal end for fixation. We also successfully performed distraction along the IM implant using the same distraction protocol as the surgical control model (**Figure R1**). In our pilot sheep study, all animals were successfully taken out for six months following the surgical procedure, demonstrating the ability to bear weight on the device and tolerate any pin tract infection risks. In summary, our additional preclinical sheep experiments have demonstrated translation relevance of our novel device. This additional experiment has been incorporated into the methods (page 15, line 3-14), results (page 7, line 30-40), discussion (page 10, line 44-45 & page 11, 1-3), reference 69 (page 21, line 3-6), and supplementary data (Page 8 and page 24, Supplementary Fig. 21).

Figure R1. Representative photos and radiographs of the sheep metatarsal bone transport model, distraction and device implantation. Photos at left column show the surgically created 30-mm bone defect in the metatarsus fix with circumferential fixators in the bone transport sheep model. The animals were treated with implantation of intramedullary implant (IMI) or without implant as blank control (BLK). The intercalary segments were transported retrogradely to the docking site

(indicated by the yellow arrows). X ray images show the affected bone fix with circumferential fixators after 15-day distraction (middle column) or 6 months after implantation (right column).

References:

1. Liu C, Zhang X, Zhang X, Li Z, Xu Y, Liu T. Bone transport with a unilateral external fixator for femoral infected nonunion after intramedullary nailing fixation: A case control study. *Medicine (Baltimore)*. 2019 May;98(20):e15612. doi: 10.1097/MD.00000000000015612. PMID: 31096468; PMCID: PMC6531196.
2. Lopez-Pliego EM, Mora-Macias J, Giraldez-Sanchez MA, Dominguez J, Reina-Romo E. Histological study of the docking site after bone transport. Temporal evolution in a sheep model. *Injury*. 2018;49(11):1987-92. Epub 2018/09/24. doi: 10.1016/j.injury.2018.09.028. PubMed PMID: 30243653.
3. Claes L, Laule J, Wenger K, Suger G, Liener U, Kinzl L. The influence of stiffness of the fixator on maturation of callus after segmental transport. *J Bone Joint Surg Br*. 2000;82(1):142-8. Epub 2000/03/04. PubMed PMID: 10697331.
4. Boyce AS, Reveal G, Scheid DK, Kaehr DM, Maar D, Watts M, Stone MB. Canine investigation of rhBMP-2, autogenous bone graft, and rhBMP-2 with autogenous bone graft for the healing of a large segmental tibial defect. *J Orthop Trauma*. 2009;23(10):685-92. Epub 2009/10/28. doi: 10.1097/BOT.0b013e3181a10378. PubMed PMID: 19858976

2. The IM implant will deliver the BMP-2 but at what rate will it biodegrade? What will be the impact of the implant on the surrounding tissue and bone formation? The implant is a potential source of infection.

Response:

We agree that an appropriate degradation is essential in the translation of this intramedullary implant. The component of this rod-shaped implant is PCL-TCP in a proportion of 4:1, which has been used in clinics (1,2). The PCL-based implants were reported to degrade completely in humans within two years (3 to 5). This slow degradation rate was also consistent with our in vivo studies (6,7), in which the degradation rates of the macroporous 80/20 PCL-TCP scaffolds at 8 weeks after implantation in the femoral head were approximately 26% in normal healthy rabbits and approximately 19% in the rabbits with corticosteroid-associated ONFH. The hydrogel coating onto the PCL-TCP rod-shaped implant is made of PEGDMA, GelMA, and alginate, and a few hundred micrometer thick, and will degrade within a few months. A 12-week subcutaneous implantation in rats reported that a PEG-based hydrogel experienced progressive degradation evidenced by significantly increased swelling ratio and decreased modulus (8). Based on linear changes between swelling ratio and modulus properties, the degradation of the hydrogels was estimated approximately 80% at 12 weeks after implantation (8). And GelMA also shows a relatively slow degradation (3.73% mass loss in 2 weeks, subcutaneously) (9). In our recent study (10), an alginate-based hydrogel was estimated to degrade at approximately 46% at 4 weeks after implantation, and 59% at 8 weeks after implantation in a cranial defect rat model, respectively. We have addressed this point in the discussion (page 9, line 13-16).

Nevertheless, we didn't find any difference in the pin tract infection rate between IM implant and surgical control in this study. In addition, we didn't observe any infection in the PCL-TCP implantation in the surgically created bone tunnel for the treatment of osteonecrosis in our rabbit

studies (6,7), and in the large defect repair studies in rats (11) and dogs (12,13). However, we acknowledge that the observation period for patients is much longer than the current animal experiment, a long-term observation or a faster degradation should be taken into consideration in the future. In summary, this animal study and previous ones applying PCL-TCP showed a relatively low degradation, but without causing any infection, indicating a relatively safe approach. We have addressed this point in the results (page 6, line 35-39).

References:

1. Markus Laubach SS, Buddhi Herath, Marie-Luise Wille, Heide Delbrück, Hatem Alabulrahman, Dietmar W. Hutmacher, Frank Hildebrand. Clinical translation of a patient-specific scaffold-guided bone regeneration concept in four cases with large long bone defects. *Journal of Orthopaedic Translation* 2022;34:73-84
2. Philipp Kobbe, Markus Laubach, Dietmar W. Hutmacher, Hatem Alabulrahman, Richard M. Sellei, and Frank Hildebrand. Convergence of scaffold-guided bone regeneration and RIA bone grafting for the treatment of a critical-sized bone defect of the femoral shaft. *Eur J Med Res* (2020) 25:70. <https://doi.org/10.1186/s40001-020-00471-w>
3. Sun H, Mei L, Song C, Cui X, Wang P. The in vivo degradation, absorption and excretion of PCL-based implant. *Biomaterials*. 2006;27(9):1735-40. Epub 2005/10/04. doi: 10.1016/j.biomaterials.2005.09.019. PubMed PMID: 16198413.
4. Cheung LK, Zheng LW. Effect of recombinant human bone morphogenetic protein-2 on mandibular distraction at different rates in an experimental model. *J Craniofac Surg*. 2006;17(1):100-8; discussion 9-10. Epub 2006/01/25. doi: 10.1097/01.scs.0000188744.06723.1f. PubMed PMID: 16432416
5. <https://www.jnjmedtech.com/en-US/product/trumatch-graft-cage-long-bone>
6. Toshiyuki Kawai, Yaser Shanjani, Saba Fazeli, Anthony W. Behn, Yaichiro Okuzu, Stuart B. Goodman, **Yunzhi P. Yang**. Customized, degradable, functionally graded scaffold for potential treatment of early stage osteonecrosis of the femoral head. *Journal of Orthopedic Research* 2018 Mar;36(3):1002-1011. doi: 10.1002/jor.23673
7. Maruyama M, Pan CC, Moeinzadeh S, Storaci HW, Guzman RA, Lui E, Ueno M, Utsunomiya T, Zhang N, Rhee C, Yao Z, Takagi M, Goodman SB, Yang YP. Effect of porosity of a functionally-graded scaffold for the treatment of corticosteroid-associated osteonecrosis of the femoral head in rabbits. *J Orthop Translat*. 2021 Mar 16;28:90-99. doi: 10.1016/j.jot.2021.01.002.
8. M. B. Browning, S. N. Cereceres, P. T. Luong, E. M. Cosgriff-Hernandez. Determination of the in vivo degradation mechanism of PEGDA hydrogels. *J Biomed Mater Res Part A*: 102A: 4244–4251, 2014
9. Heltmann-Meyer S, Steiner D, Müller C, Schneidereit D, Friedrich O, Salehi S, Engel FB, Arkudas A, Horch RE. Gelatin methacryloyl is a slow degrading material allowing vascularization and long-term use in vivo. *Biomed Mater*. 2021 Sep 6;16(6). doi: 10.1088/1748-605X/ac1e9d. PMID: 34406979.
10. Youngbum Park, Sien Lin, Yan Bai, Seyedsina Moeinzadeh, Sungwoo Kim, Jianping Huang², Uilyong Lee⁴, Ngan Huang, **Yunzhi Peter Yang**. Controlled sequential release of BMP-2 and IGF-1 in microparticles dual delivery system promote calvarial bone defect healing. *Tissue Engineering A*. 2022 Mar 31. doi: 10.1089/ten.TEA.2022.0002
11. Toshiyuki Kawai, MD PhD, Chi-chun Pan, PhD, Yaichiro Okuzu, Takayoshi Shimizu,

- Alexander M. Stahl, Shuichi Matsuda, William J. Maloney, **Yunzhi P. Yang**. Combining a vascular bundle and 3D printed scaffold with BMP2 improves bone repair and angiogenesis in critical size bone defect in rats. *Tissue Engineering Part A* Volume 27, Numbers 23 and 24, 2021; 1517-1525
12. **Yunzhi Peter Yang**, Kevin M. Labus, Benjamin C. Gadowski, Arnaud Bruyas, Jeremiah Easley, Brad Nelson, Ross Palmer, Kirk McGilvray, Daniel Regan, Christian M. Puttlitz, Alexander Stahl, Elaine Lui, Jiannan Li, Seyedsina Moeinzadeh, Sungwoo Kim, William Maloney, Michael J. Gardner. Osteoinductive 3D Printed Scaffold healed 5 cm Segmental Bone Defects in the Ovine Metatarsus. *Scientific Reports* **11**, 6704 (2021). <https://doi.org/10.1038/s41598-021-86210-5>
13. **Yunzhi Peter Yang**, Benjamin C. Gadowski, Arnaud Bruyas, Jeremiah Easley, Kevin M. Labus, Brad Nelson, Ross Palmer, Kirk McGilvray, Christian M. Puttlitz, Dan Regan, Alexander Stahl, Elaine Lui, Jiannan Li, Seyedsina Moeinzadeh, Sungwoo Kim, William Maloney, Michael J. Gardner. Investigation of a Prevascularized Bone Graft for Large Defects in the Ovine Tibia. *Tissue Engineering Part A*, 2021 Dec;27(23-24):1458-1469. doi: 10.1089/ten.TEA.2020.0347. Epub 2021 Jun 11.

3. The authors chose CT scan results as the outcome measure. In most bone healing studies of this nature biomechanical testing is used as the outcome measure. It is important to determine the quality of the bone that is present at both the distal docking site and the proximal regenerate. Furthermore, the study appears to be underpowered.

Response:

In this study, we have performed three-point bending test on the femurs at post operation day 34 (POD34) or 55 (POD55). We agree that it would be better to compare the mechanical properties of the distal docking site and proximal regenerate, respectively. In this study, we found that almost all the healed specimens were broken at the docking site when reached to their ultimate load. Furthermore, non-union or delayed union could be only found at the docking site, which is consistent with clinical evidence, indicating a main bottle neck for bone transport. In this study, we also showed animal gait before and after fixator removal, which may suggest load bearing potential after the treatments. We have supplemented more information in the results of mechanical test (page 5, line 23-28).

For the sample size estimation in this study, we basically used maximum load in our recent study as the primary outcome using G-Power calculator. As there is no published mechanical data of rat bone transport model, we used previous data in a rat limb distraction osteogenesis model (Pan Q, et al., 2021) for the sample size estimation. As shown in the following Figure R2, when the mean and SD of control group and experimental group were given, we can have an effect size d equal to 1.8031. When the power set to 0.9 and α error set to 0.05, the sample size would be 8. We understand that the bone transport and bone lengthening animal models are similar but not the same, a larger sample size could be better in terms of increasing the power in this animal model. The information can be found in the Statistical analysis.

Figure R2. A priori power analysis (G*Power, Universität Düsseldorf) based on the previous studies using a similar rat bone lengthening DO model. When the mean and SD of control group and experimental group were given, an effect size d equals to 1.8031. When the power set to 0.9 and α error set to 0.05, the sample size would be 8.

References:

1. Pan Q, Li Y, Li Y, Wang H, Kong L, Yang Z, Zhang X, Bai S, Zong Z, Chen G, Lin S*, Li G*. Local administration of allogeneic or autologous bone marrow derived mesenchymal stem cells enhanced bone formation similarly in distraction osteogenesis. *Cytotherapy*, 2021 Jul;23(7):590-598. Sdoi: 10.1016/j.jcyt.2020.12.005.

4. It is not clear from the data presented how many bone defects actually healed on plain radiographs or even CT scans with 3D reconstruction.

Response:

X-ray imaging has been conducted for each rat in a weekly basis since the date of operation until sacrificed. And micro-CT analysis has also been conducted for each specimen. As shown in the result of micro-CT analysis, the union rates have been presented in Figure 2C & 2D. For example, the union rate on POD34 or POD55 in the BLK group was 12.5% or 25%, respectively. It means that only 1/8 or 2/8 of the bone defects healed on POD34 or POD55, respectively.

We agree with the reviewer that observative data using X-ray imaging results could be helpful. We have supplemented results of X-ray imaging (Figure R3 as follows) to show the dynamic bone healing in live animal at POD5 (before distraction), POD13 (after distraction), POD34 (3-week consolidation), and POD55 (6-week consolidation) in the Supplementary Fig. 7. However, the resolution of X-ray images was not that high enough to distinguish the bridging at the docking site.

That's why we determined the union rate by using micro-CT images.

Figure R3. X ray results of dynamic bone healing in rat. The X-ray imaging was conducted on POD5 (before distraction), POD13 (after distraction), POD34 (3-week consolidation), and POD55 (6-week consolidation). White and red arrows indicate the docking site and regenerate site, respectively.

5. It appears that BMP-2 is secreted for approximately 21 days. This may not be a sufficient duration of production in trying to heal large segmental defect in humans.

Response:

We understand the concern from reviewer. In this proof-of-concept study, we used a bioactive implant that achieved a 21-day release kinetics and significantly improved the bone repair in both the regenerate and docking sites in this rat bone transport model, in which we created a 4-mm transportable segment and a 8-mm large bone defect. For healing a large segmental defect in humans, a prolonged release kinetics of BMP2 is required. We have developed a strategy to further prolong the release kinetics of BMP2 by coating additional hydrophobic PCL layer(s) onto the HyTEC implant. Figure R4 shows the prolonged release using such strategy. The BMP2-laden HyTEC graft is coated with a single layer or triple layers of PCL and make modified HyTEC graft (mHyTEC). The concentration of PCL solution and the number of deposited PCL layers could be changed to tune the physical characteristics of mHyTEC grafts and release kinetics of proteins. As shown in Figure R4, while 92% of the encapsulated model protein, bone serum albumin (BSA) was released from BSA-laden HyTEC without coating after 14 days, the rate of BSA release was reduced to 92% after 70 days by mHyTEC (1L) with addition of 1 layer of hydrophobic barrier PCL, or 80% in 91 days by mHyTEC (3L) with addition of 3 layers of hydrophobic barrier PCL (**Figure R4a**). More encouragingly, the amount of released BMP2 from the BMP2-laden mHyTECs after 28 days in PBS decreased from 84% to 62% by mHyTEC (1L) or 24% by mHyTEC (3L) (**Figure R4b**). As such, we will apply the same mHyTEC strategy for achieving a prolonged release of BMP2 in a clinically relevant sheep model and humans in the future. We have supplemented more information in the methods (page 12, line 18-21), results (page 3, line 40-44 & page 4, line 1-2, Supplementary Fig. 4, and discussion (page 9, line 9-13).

Figure R4. Prolonged release kinetics of BSA and BMP2 from the rod-shaped mHyTEC grafts. (a) Release kinetics of BSA from the HyTEC and mHyTEC grafts with one or three PCL layers for 91 days or 13 weeks. (b) Release kinetics of BMP2 from the HyTEC and mHyTEC grafts with one or three PCL layers for 28 days. Data are shown as mean and SD (n = 3).

6. In a human, how would the authors stabilize the intramedullary implant? If there is micromotion of the implant this could impact bone healing.

Response:

As shown in the responses to the first comment above, an additional sheep experiment has been conducted to address this important concern. First, as shown in our sheep pilot study, the intramedullary implant can be stabilized by using a 0.6mm k-wire through the implant on distal or proximal end for fixation when applied with circumferential fixator. In addition, the animal was successfully taken out for six months following the surgical procedure, demonstrating the stability of the device by k-wrie fixation and the ability to bear weight on the device. Second, the shape of implant itself can be custom-designed for each patient to fit into the patient’s bone marrow cavity based on the CT results. And the rigidity of implant can be enhanced by optimization of proportion of PCL and TCP, and porosity of scaffold. The additional experiment has been incorporated into the methods (page 15, line 3-14), results (page 7, line 30-40), discussion (page 10, line 44-45 & page 11, 1-3), reference 69 (page 21, line 3-6), and supplementary data (Page 8 and page 24, Supplementary Fig. 21).

7. A Table delineating each group and the various tests done, etc. would be helpful to the reader.

Response:

A Table delineating each group and the various tests has been provided in the Supplementary Materials (Table S1) and results of the revised manuscript (page 4, line 7-8).

Table S1. Animal groups, treatments and the assessments in the study using rat bone transport model

Groups	Treatments (treated with bone transport)	Sample size & timepoints	X-ray, μ CT, 3-point bending (n = 8)	Histomorphometry (n = 3)	Histology/ Immunohistochemistry (n = 5)	Gait (n = 2)

BLK	Blank (BLK) control, on implant	n = 8 (POD34) + 8 (POD55)	√	√	√	
CS + B2	Collagen sponge with BMP-2 (2 μg) at defect site	n = 8 (POD34) + 8 (POD55)	√	√	√	
IMI	IM implant (IMI) only, no BMP-2	n = 8 (POD34) + 8 (POD55)	√	√	√	√
IMI + B0.5	IMI + BMP-2 (0.5 μg)	n = 8 (POD34) + 8 (POD55)	√	√	√	
IMI + B2	IMI + BMP-2 (2 μg)	n = 8 (POD34) + 8 (POD55)	√	√	√	√
IMI + B6	IMI + BMP-2 (6 μg)	n = 8 (POD34)	√	√	√	

Reviewer #2 (Remarks to the Author):

The manuscript by Yang and colleagues under consideration at Nature Communications describes a tissue engineering approach for sustained delivery of BMP-2 for bone regeneration. This IM implant model and other procedures would benefit from controlled release of bone repair morphogens. This is a carefully designed translational study with considerable evidence for successful procedure with enhanced osseointegration. The following points should be taken into consideration in a revised manuscript:

1) The rationale for the complex coating procedure is not well laid out. Why would you need GelMA then alginate followed by GelMA and PEGDMA after freeze drying? Some description of the role for mediating delivery, coating stability, etc. is needed in the results and discussion.

Response:

We thank the reviewer for the opportunity to allow us to clarify the rationale of the multiple steps. Basically, the surface of PCL-TCP filament is hydrophobic, making it difficult for hydrogel coating. To overcome this challenge, we developed a method to fabricate a thick hydrogel coating onto the hydrophobic PCL-TCP filament. The method comprises consecutive steps to increase hydrophilicity, improve hydrogel adhesion, and stimulate surface-initiated crosslinking. As shown in the revised Figure 1 in the manuscript, the PCL-TCP filament was first treated with alkaline immersion and freeze-thaw cycle, which creates carboxylic groups (-COOH) and increases surface area. Then the first thin layer of GelMA is coated onto the alkaline treated PCL-TCP filament and presents double bonds on the surface that will form covalent binding with the GelMA and PEGDMA within the hydrogel coating. After that, CaSO₄ is deposited onto the GelMA coated PCL-TCP filament. When the treated filament is dipped into a solution of alginate, gelatin methacrylate (GelMA), poly(ethylene glycol) dimethacrylate (PEGDMA), and BMP2, a layer of soft gel forms on the surface due to the diffusion of calcium ions from the surface that physically crosslink alginate (Figure 1A). The BMP2 laden hydrogel layer is then stabilized via covalent crosslinking of GelMA and PEGDMA via light exposure. Due to a surface initiated crosslinking mechanism, the thickness of the hydrogel layer on the surface of PCL-TCP scaffolds can be controlled, and a large quantity of hydrogel coating could be achieved (Figure 1C-1G). In summary, the freeze-dried or rehydrated hydrogel coating comprises a tough interpenetrating network of the hydrogel made of physically crosslinked alginate and covalently crosslinked GelMA/PEGDMA, and creates strong binding to the surface of PCL/ β -TCP due to a combination of covalent binding of the hydrogel to the functional groups on the PCL/ β -TCP surface and mechanical interlocking between the hydrogel macromonomers and cracked surface of the treated PCL-TCP. This strong binding ensures the dried or rehydrated hydrogel coating is not easily peeled off during or after implantation. We have supplemented more information in the results (page 3, line 33-37), Figure 1 (page 22), discussion (page 8, 41-45 & page 9, 1-6), Supplementary Materials (Supplementary Fig. 1D-1F & 2).

2) The data indicating dynamic mineralisation at day POD34 is compelling for the IMI+Bx groups. However, it is not clear why there is no significant difference at POD55. This should be discussed.

Response:

The results can be explained by the release kinetic of BMP-2. In this study, the BMP-2 could be released for 21 days *in vitro* as shown by the release kinetic study. We believe that the sustained release of BMP-2 could promote mineralization at the regenerate and docking sites in a period as

long as 21 days. However, when all the BMP-2 was released afterwards, there could be no additional BMP-2 locally to maintain the mineralization, indicating bone formation was still active at the early 3 weeks of consolidation, but gradually became less active afterwards. Then an enhanced effect on mineralization could be found at early phase like POD34, but not at later phase like POD55. This response has been incorporated into the discussion part (page 9, line 35-40).

3) Evidence for increased osteogenesis and decreased rate of infection is clear. However, one of the potential caveats of integrating a soft/composite layer at the interface of a hard biomaterial is the potential for a weak interface. Some evidence that the new coating has at least comparable long term adhesion to the bone is important.

Response:

The interface between soft hydrogel and PCL-TCP rigid scaffold is a robust integration. As what we have responded to the first comment of the reviewer, the freeze-dried or rehydrated hydrogel coating comprises a tough interpenetrating network of the hydrogel made of physically crosslinked alginate and covalently crosslinked GelMA/PEGDMA, and creates strong binding to the surface of PCL-TCP due to a combination of covalent binding of the hydrogel to the functional groups on the PCL-TCP surface and mechanical interlocking between the hydrogel macromonomers and cracked surface of the treated PCL-TCP. This strong binding ensures the dried or rehydrated hydrogel coating is not easily peeled off during or after implantation. We understand that the insertion of the IM implant into the marrow cavity and the movement of the transportable bone segment along the IM implant will generate frictional forces and shear stresses onto the hydrogel layer of the IM implant. As such, we applied a preemptive approach. The diameter of our IM implant is slightly smaller than the inner diameter of IM cavity, leaving a space between our IM implant and the relatively rigid inner cancellous bone to avoid direct contact. Additionally, the strong binding between the hydrogel layer and the PCL-TCP rod retains the integrity of the hydrogel layer. What's more, additional characterization on the interface of PCL-TCP and the hydrogel coating has been done during this revision (**Fig. R5-R7**). Revisions have been made in the methods (page 12, line 12-15; supplemental methods (page 4 & 5), results (page 3, line 31-35), Fig. 1F & 1G, Supplementary Figure 1-3), discussion (page 8, 39-45 & page 9, 1-6).

Note, this hydrogel coated rod-shaped IM implant was placed into the marrow cavity and intended to be used as a carrier for controlled sustained release of BMP2. The mineralization of the regenerating bone in the bone defect will be expected after the completement of distraction. Our *in vivo* studies have shown convincing results in bone healing through the sustained release of BMP2 from the IM implant delivery in the bone transport model.

Fig. R5. Microphotographs of PCL-TCP filament surface and the interface between hydrogel coating and PCL-TCP filament. (E) Cross-sectional or (F) longitudinal hydrogel-coated PCL-TCP filaments, showing a tight adhesion and smooth hydrogel-filament interface. Inserted images showing higher magnifications.

Fig. R6. Contact angle goniometry at the filament-coating interface. (A) Water droplet (2 μ L) centered at the interface of the PCL-TCP filament and Hytec gel coating. (B) Contact angle measurements of the filament and gel. Data are represented as a bar graph with mean and s.e.m. values (n = 10).

Fig. R7. A customized 3D printed PCL-TCP device for evaluating the adhesion of the hydrogel-coated PCL-TCP scaffold made by HyTEC technique. (A) A schematic of the 3D printed PCL-TCP device which was used to measure the adhesion of freeze-dried composite hydrogels to scaffolds. The device composed of two concentric cylinders connected through two bridges. (B) A schematic of the shear test experiment with cross sectional view of the device. Freeze-dried hydrogel was filled between the inner and outer ring. The top arrow represents the force applied to the device during the test. (C) The 3D printed device before loading hydrogel. (D) The device with freeze-dried hydrogel within the gap

between two concentric cylinders. (F) Experimental set up to measure interfacial shear strength. (E) Six samples were tested, where three samples were observed to break at the inner ring interface, and three samples were observed to break at the outer ring interface. The overall interfacial shear strength of 0.609 ± 0.194 MPa was obtained by considering combined inner and outer interfacial strength measurements. Data are represented as a bar graph with mean and s.e.m. values (n = 6).

4) Another complication could be shelled particles that might trigger inflammation and potentially bone resorption. Considering the increase in osteoclasts, I am wondering if this could lead to bone resorption around the structure at later time points.

Response:

In this study, the increase in osteoclasts caused by BMP-2 and the bone resorption triggered by shelled particles was not observed. However, further study will be conducted to see longer term effect of our IM implant.

As shown in our results (Figure 5B & 5D), bone resorption was investigated in this study. We found higher number of osteoclasts per bone surface (Oc.N/BS) at the docking sites in the (IMI + B2) and (IMI + B6) groups on POD34 as stained by TRAP assay, which is consistent with the increased expression of osteogenic marker (Osteocalcin) and angiogenic marker (VEGF). Such the coupling between osteoblasts and osteoclasts suggests robust bone formation and early remodeling. However, we did not see a significant difference in the osteoclastic activity at the regenerate sites among the groups on POD34 (Fig. 5B & Supplementary Fig. 12). In addition, compared to BLK and CS+B2 groups, IMI and (IMI+BMP2) groups did not increase the number of osteoclasts on POD55. As such, the increased number of osteoclasts is probably the result of BMP-2 induced robust bone formation instead of the shelled particles. However, further study will be conducted to determine the potential shelled particles effect in a longer term. This point has been addressed in the results (page 6, line 4-6, 15).

5) Page 8 Line 17: “Coating of the surface with GelMA presented double bonds on the surface for covalent linking to the hydrogel.” This statement is confusing. Are the authors referring to pendant acryloyl moieties for further modification? This needs to be reworded with the surface chemistry explicitly defined. If the authors have performed rigorous characterization of the materials interface in previous work, this should be discussed in more detail. Alternatively, more details of the interfaces between the multiple materials is important for the broad readership.

Response:

That is correct. The pendant acryloyl moieties of GelMA onto the surface provide a covalent binding site with the GelMA and PEGDMA in the hydrogel. The rationale behind multiple steps of hydrogel coating has been updated in discussion (page 8, 39-45 & page 9, 1-6) as in the response to Comment 1 and 3. We have elaborated the characterization of hydrogel coating and the rationale behind in the revised manuscript. An additional characterization on the interface of PCL-TCP and the hydrogel coating has been done during this revision as shown above in the response to Comment 3.

6) HepMA incorporation is described in the discussion but not in the results. More details are required for describing and evaluating the soft materials that facilitate sustained BMP release.

Response:

The release of BMP2 from GelMA/PEGDMA hydrogel is mainly controlled by the molecular diffusion from the porous hydrogel network, and enzymatic degradation of the GelMA. HepMA was added to GelMA/PEGDMA hydrogel to slow down the release kinetics of BMP2 from the GelMA/PEGDMA hydrogel. Heparin is a negatively charged glycosaminoglycan that has affinity for many growth factors including BMP-2 through electrostatic interactions (1-3). It has been reported that the presence of HepMA in hydrogels slows down the release kinetics of BMP-2 (4).

The rationale of using HepMA has been added into discussion (page 8, line 41-44) and the reference 54 (page 20, line 6-8) has been added.

References:

1. Zhang F, Zheng L, Cheng S, Peng Y, Fu L, Zhang X, Linhardt RJ. Comparison of the Interactions of Different Growth Factors and Glycosaminoglycans. *Molecules*. 2019 Sep 16;24(18):3360. doi: 10.3390/molecules24183360. PMID: 31527407; PMCID: PMC6767211.
2. Hettiaratchi MH, Krishnan L, Rouse T, Chou C, McDevitt TC, Guldborg RE. Heparin-mediated delivery of bone morphogenetic protein-2 improves spatial localization of bone regeneration. *Sci Adv*. 2020 Jan 3;6(1):eaay1240. doi: 10.1126/sciadv.aay1240. PMID: 31922007; PMCID: PMC6941907.
3. Joung YK, Bae JW, Park KD. Controlled release of heparin-binding growth factors using heparin-containing particulate systems for tissue regeneration. *Expert Opin Drug Deliv*. 2008 Nov;5(11):1173-84. doi: 10.1517/17425240802431811. PMID: 18976129.
4. Jeon O, Powell C, Solorio LD, Krebs MD, Alsberg E. Affinity-based growth factor delivery using biodegradable, photocrosslinked heparin-alginate hydrogels. *J Control Release*. 2011 Sep 25;154(3):258-66. doi: 10.1016/j.jconrel.2011.06.027. Epub 2011 Jul 2. PMID: 21745508; PMCID: PMC3541683.

Minor:

Page 4 Line 13 – Extend should read Extent

Page 4 line 24 – compared should read comparison

Response:

Revised accordingly in page 4, line 20 and line 30

REVIEWERS' COMMENTS

Reviewer #2 (Remarks to the Author):

The authors have considered my comments and responded adequately with revised text and figures. In my opinion the manuscript is ready for publication.

Reviewer #3 (Remarks to the Author):

The authors have largely addressed the issues raised by the Reviewers.

Some points need to be clarified

1. it is possible that the manuscript has not been revised by the English speaking co-authors: several idiosyncrasies in language need to be addressed.
2. 'Ilizarov technique, also knows as distraction osteogenesis.'. This is totally incorrect. Please change to 'Ilizarov technique, based on the principles of distraction osteogenesis,'
3. line 46: 'this procedure starts with an osteotomy' change to 'this procedure starts with a low energy osteotomy'
4. line 47: 'Ilizarov technique includes two categories:acute shortening and lengthening and bone transport' please change to 'Ilizarov described two techniques:acute shortening and subsequent lengthening,and bone transport'
5. line 91 and throughout the manuscript. I question whether this is really 'translational research'. The authors have described how the results of their investigation in a small animal model can be applied to a larger animal model, but there is no evidence that their results can be translated to humans. In this respect, these are still animal studies

RESPONSES TO REFEREES

Reviewer #2 (Remarks to the Author):

The authors have considered my comments and responded adequately with revised text and figures. In my opinion the manuscript is ready for publication.

Response: We sincerely thank the reviewer for the comments.

Reviewer #3 (Remarks to the Author):

The authors have largely addressed the issues raised by the Reviewers.

Some points need to be clarified

1. it is possible that the manuscript has not been revised by the English speaking co-authors: several idiosyncrasies in language need to be addressed.

Response: We sincerely appreciate the comments from the reviewer. Native English speakers and orthopaedic surgeons Dr. Michael Gardner and Dr. David Mohler have been invited to help revise this manuscript.

2. 'Ilizarov technique, also knows as distraction osteogenesis.'. This is totally incorrect. Please change to 'Ilizarov technique, based on the principles of distraction osteogenesis,'

Response: Revised accordingly in the introduction (line 5 & 6, page 2).

Corrected sentence: "In particular, the Ilizarov technique, based on the principles of distraction osteogenesis (DO), represents a well-established limb salvage procedure for managing large bone defects."

3. line 46: 'this procedure starts with an osteotomy' change to 'this procedure starts with a low energy osteotomy'

Response: Revised accordingly in the introduction (line 7, page 2).

Corrected sentence: "This procedure starts with a low energy osteotomy, followed by lengthening over an external or internal flexible fixator."

4. line 47: 'Ilizarov technique includes two categories: acute shortening and lengthening and bone transport' please change to 'Ilizarov described two techniques: acute shortening and subsequent lengthening, and bone transport'

Response: Revised accordingly in the introduction (line 9, page 2).

Corrected sentence: "Ilizarov described two distinct techniques: acute shortening and subsequent lengthening, and bone transport."

5. line 91 and throughout the manuscript. I question whether this is really 'translational research'. The authors have described how the results of their investigation in a small animal model can be applied to a larger animal model, but there is no evidence that their results can be translated to humans. In this respect, these are still animal studies

Response: Thank you for your question, and we appreciate your feedback.

Our team is dedicated to translational research by taking step-by-step approaches, progressing from small animal models to large animals, and ultimately aiming for clinical applications. As you can see, our team comprises experienced surgeons and physician scientists who play crucial roles in our research efforts. Together, we are dedicated to making

progress towards future patient treatments.

However, we do acknowledge that the animal experiments are still in a preliminary stage, and clinical trials are necessary to further assess the translational potential of the IM implant device. To accurately reflect the meaning of our study, we have made revisions to some relevant statements in the manuscript. For instance, we have revised the statement "To demonstrate the translational potential of the HyTEC IM implant technique" (line 31, page 7) to "To demonstrate the feasibility of the HyTEC IM implant technique in large animals."